

**Geosphere Coupling and Hydrothermal Anomalies before the 2009 Mw 6.3**
**L'Aquila Earthquake in Italy**
L.X. Wu[1,5*], S. Zheng[2], A. De Santis[3], K. Qin[1], R. Di Mauro[4], S.J. Liu[5] and M. L.
Rainone[4]
*1 School of Environment Science and Spatial Informatics, China University of Mining*
*and Technology, Xuzhou, China*
*2 Academy of Disaster Reduction & Emergency Management, Beijing Normal*
*University, Beijing, China*
*3 Istituto Nazionale di Geofisica e Vulcanologia, Sezione Roma 2, Roma, Italy*
*4 Dipartimento di Ingegneria e Geologia, Chieti University, V. Vestini 31, 66013*
*Chieti Scalo, Italy*
*5 Northeast University, Shenyang, China*
*Corresponding author: Lixin Wu,
School of Environment Science and Spatial Informatics, China University of Mining
and Technology, Xuzhou, China;
Email: awulixin@263.net, wlx@cumt.edu.cn
**Abstract:** The earthquake (EQ) anomalies associated with the April 6, 2009 Mw 6.3
L'Aquila EQ have been widely reported. Nevertheless, the reported anomalies have
not been so far synergically analyzed to interpret or prove the potential LCA coupling
process. Previous studies on *b*-value are also insufficient. In this work, the
spatio-temporal evolution of several hydrothermal parameters related to the
coversphere and atmosphere, including soil moisture, soil temperature, near-surface
air temperature, and precipitable water, was comprehensively investigated. Air
temperature and atmospheric aerosol were also statistically analyzed in time series
with ground observations. An abnormal enhancement of aerosol occurred on March
30, 2009 and thus proved quasi-synchronous anomalies among the hydrothermal
parameters from March 29 to 31 in particular places geo-related to tectonic thrusts and
local topography. The three-dimensional (3D) visualization analysis of *b*-value
revealed that regional stress accumulated to a high level, particularly in the L'Aquila
basin and around regional large thrusts. Finally, the coupling effects of geospheres
were discussed, and a conceptual LCA coupling mode was proposed to interpret the
possible mechanisms of the multiple quasi-synchronous anomalies preceding the
L'Aquila EQ. Results indicate that $CO_2$-rich fluids in deep crust might have played a
significant role in the local LCA coupling process.



## 1. Introduction

The thermal anomalies occurring before large and hazardous earthquakes (EQs) have been extensively observed from satellites or on the Earth's surface. In particular, several thermal parameters, including thermal infrared radiation (TIR) [Tronin et al., 2002; Saraf and Choudhury, 2004], surface latent heat flux [Dey and Singh, 2003; Qin et al., 2012, 2014a], and outgoing longwave radiation [Ouzounov et al., 2007; Jing et al., 2012], have been proven to be related to tectonic seismic activities. With the development of Earth observation technologies and anomaly recognition methods [e.g., Wu et al., 2012; Qin et al., 2013], non-thermal anomalous variations in geochemical and electromagnetic signals from different spheres of the Earth may indicate complex geosphere coupling effects during the slow preparation phase of EQs. During the past decades, several mechanisms or hypotheses for interpreting thermal anomalies have been proposed; examples include the positive hole (P-hole) effect [Freund, 2011], transient electric field [Liperovsky et al., 2008], frictional heat of faults [Geng et al., 1998; Wu et al., 2006], and the greenhouse effect caused by Earth degassing [Tronin et al., 2002]. A unified lithosphere–atmosphere–ionosphere coupling model was proposed to explain the inherent links among different parameters [Liperovsky et al., 2008a; Pulinets and Ouzounov, 2011; Pulinets, 2012]. This model has been verified by several case studies on the spatio-temporal features of the anomalies of multiple parameters [Pulinets et al., 2006; Zheng et al., 2014]. Wu et al. [2012] emphasized not only the effect of the coversphere (including water bodies, soil/sand layers, deserts, and vegetation on the Earth's surface) on pre-EQ anomalies but also the importance of this transition layer from the lithosphere to the atmosphere. The coversphere performs the vital functions of producing observable signals and enlarging or reducing the transmission of electric, magnetic, electromagnetic, and thermal signals from the lithosphere to the atmosphere, and even to satellite sensors. Although the existence of many diagnostic precursors,such as crustal strain, seismic velocity, hydrological change, gas emission and electromagnetic signals, and their usefulness for earthquake forecasting is still



controversial [Cicerone et al 2009; Jordan et al 2011]. With the abundant data such
provided by Global Earth Observation System of System (GEOSS), multiple
parameters from the integrated Earth observation should be encouraged to test for
earthquake anomaly recognition and advance knowledge of precursor signals. The
2009 Mw6.3 L'Aquila EQ may provide an ideal opportunity for us to further cognize
various change of observational signals in geosphere system and understand their
possible link with geophysical survey.
A Mw 6.3 EQ struck the Abruzzi region in central Italy on April 6, 2009 (01:32 UTC),
and its epicenter was located at 42.34 °N/13.38 °E (depth of 9.5 km), which was near
the city of L'Aquila (Fig.1). According to the Istituto Nazionale di Geofisica e
Vulcanologia (INGV), many strong foreshocks had been occurring since December
2008, and more than 10,000 aftershocks had been recorded until September 2010.
Previous geological studies stated that the present-day geologic setting along the
Italian peninsula related to the N-S convergence zone between the African and the
Eurasian plates is particularly complex because different processes occur
simultaneously and in close proximity [Montone et al., 2004; Galadini et al., 2000].
Central Italy experiences active NE-SW extensional tectonics approximately
perpendicular to the Apenninic fold and thrust belt [Montone et al., 2012]; a city in
this region is L'Aquila, which is bounded by the *Olevano–Antrodoco* and *Gran Sasso*
thrusts at the west and north sides, respectively. In 2009, the L'Aquila main shock
occurred as a result of normal faulting (*Paganica* fault, PF) and as a primary response
to the Tyrrhenian basin opening faster than the compression between the Eurasian and
African plates [USGS, 2009].
A large number of the precursory anomalies of the 2009 L'Aquila EQ were reported
after the main shock. These anomalous parameters included thermal properties,
electric and magnetic fields, gas emissions, and seismicity [Akhoondzadeh et al., 2010;
Biagi et al., 2009; Bonfanti et al., 2012; Cianchini et al., 2012; De Santis et al., 2011;
Eftaxias et al., 2009; Genzano et al., 2009; Gregori et al., 2010; Lisi et al., 2010;
Papadopoulos et al., 2010; Piroddi and Ranieri, 2012; Plastino et al., 2010; Pulinets et
al., 2010; Rozhnoi et al., 2009; Tsolis and Xenos, 2010]. Many of the existing reports



revealed the existence of temporal quasi-synchronism among the several anomalies of
different parameters related to different geospheres (Table 1). We believe that the
geosphere coupling effects could support or interpret the occurrence of the various
precursory anomalies of the 2009 L'Aquila EQ. Moreover, we hypothesize the
possible role of the coversphere in the process of lithosphere–coversphere–
atmosphere (LCA) coupling, in which the radiation transmission caused anomalous
thermal infrared signals in satellite sensors.
Air ionization and ion hydration are generally known as critical physical processes
that result in different types of EQ precursors between the ground surface and the
lower atmosphere [Pulinets and Ouzounov, 2011; Freund, 2011]. However, a
corresponding observation of complementary parameters related to the coversphere,
such as humidity, water vapor, heart flux, and atmospheric aerosol, is not
comprehensive enough to obtain a plain validation.
The seismic $b$-value describes the fundamental relationship between the frequency
and the magnitude of EQs, which is known as the Gutenberg–Richter law [Gutenberg
and Richter, 1944], and is widely applied in tectonic seismicity studies. The $b$-value
represents the size distribution of abundant seismic events of small to moderate
magnitudes; it is associated with several physical properties, such as regional stress,
material homogeneity, and temperature gradient [Gulia and Wiemer, 2010; Mogi,
1962; Schorlemmer et al., 2005; Schorlemmer and Wiemer, 2004, 2005; Tormann et
al., 2015; Urbancic et al., 1992; Warren and Latham, 1970; Wiemer and Wyss, 2002;
Wyss and Wiemer, 2000]. Hence, the $b$-value is possibly a proxy of crust stress
conditions and could therefore act as a crude stress meter for seismicity observed in
the lithosphere [Tormann et al., 2014]. Although the time sequence of the $b$-value
based on microseismicity data before and after the 2009 L'Aquila EQ has been
analyzed and has revealed the quasi-synchronous features of the $b$-value relative to
other parameters [De Santis et al., 2011], the correlations of various anomalies in the
coversphere and lithosphere remain unclear because of the absence of essential
geospatial analysis. Moreover, various factors directly influence the thermal radiation
signals observed by satellite sensors; these factors include atmosphere properties



(absorption, scattering, and emission of water vapor, as well as aerosol particles),
thermal condition of the Earth's surface (meteorological condition, soil moisture and
components, vegetation cover, and surface roughness), and the complex thermal
process of geo-objects. In view of remote sensing physics and the LCA coupling
effect, we have reason to believe that other parameters characterized by the
above-mentioned factors in relation to the coversphere and atmosphere should have
presented temporal quasi-synchronism and spatial consistency with the reported
thermal anomalies before the main shock of the 2009 L'Aquila EQ.
Several hydrothermal parameters related to the coversphere and atmosphere,
including soil water and temperature, precipitable water, air temperature, and
atmospheric aerosol, are comprehensively analyzed in this study to explore the
possible coupling effects preceding the 2009 Mw 6.3 L'Aquila EQ. The 3D dynamic
evolution of the *b*-value is also analyzed to further investigate the potential
correlations of multiple parameter anomalies related to the coversphere and the
dynamics of the lithosphere. Furthermore, the variation of some parameters after the
main shock is analyzed for comparison. From retrospective analyses of data collected
prior to this earthquake, we finally attempt to discuss the geosphere coupling process
and propose a model for interpreting the coupling effects with the support of previous
geophysical researches.
**2. Analysis of hydrothermal parameters**
**2.1 Data and method**
Four parameters related to the coversphere and atmosphere, namely, volumetric soil
moisture level 1 (SML1) at 0–7 cm below ground level, soil temperature level 1
(STL1) at 0–7 cm below ground level, near-surface air temperature at a height of 2 m
(TMP2m), and precipitable water of the entire atmosphere column (PWATclm), were
analyzed in long-term intervals and within two months before and after the main
shock. The six-hourly values of the SML1 and STL1 parameters were 00:00, 06:00,
12:00, and 18:00 every day according to ERA-Interim, which is a series of the latest
global atmospheric reanalysis products produced by the European Centre for



Medium-Range Weather Forecasts to replace the ERA-40. The gridded data were
transformed into a regular 512 ° longitude by 256 ° latitude N128 Gaussian grid with
0.71 ° × 0.71 ° spatial resolution (http://apps.ecmwf.int/datasets/data). The TMP2m and
PWATclm datasets also comprised six-hourly values based on the Final (FNL)
Operational Global Analysis system of the National Center for Environmental
Prediction (NCEP), which was produced with the same NCEP model as that used for
the Global Forecast System (http://rda.ucar.edu/datasets). The NCEP-FNL data were
also represented in a Gaussian grid with 1 ° × 1 ° spatial resolution (360 ° longitude by
181 ° latitude). All the data from March and April 2000-2009 were investigated. The
datasets containing information on the air temperature and aerosol optical depth
(AOD) from ground-based observations were considered and compared with the
results from the assimilation data to verify the key coupling process of the anomalies.
The air temperature data were obtained from the L'Aquila weather station
(42.22 °N/13.21 °E, elevation of 680 m, shown as yellow circle in Fig. 1), whereas the
AOD data were obtained from the Roma station (41.84 °N/12.65 °E, elevation of 130 m,
shown as yellow triangle in Fig. 1) of the Aerosol Robotic Networks (AERONET,
http://aeronet.gsfc.nasa.gov/). With respect to the epicenter, the Roma station, which
uses the Cimel Electronique CE318 sunphotometer to measure aerosol optical
properties, is the only nearby station with available data.
First, we analyzed the long time series of the SML1, STL1, TMP2m, and PWATclm
data on the epicenter pixel (42.34 °N, 13.38 °E, shown as black rectangular boxes in
Figs. 2.2-2.5). To compare the data in 2009 with historical data, the mean ($\mu$) and
standard deviation ($\sigma$) were calculated using data from multiple years (2000–2008).
Here,an deviation with overquantity more than $\mu$+1.5$\sigma$ threshold was defined as an
alternative anomaly for each parameter on the epicenter pixel. For confutation
analysis, we also compared the 2009 data with the data from 2006 (green line in Fig.
2), which is regarded as a silent year for its seismicity rate (≤10 EQs with M3+
according to the INGV catalog for this area). After processing the preliminary data
and checking for errors, we found that the anomalies of multiple parameters were
more remarkable at 06:00 UTC than in other periods. Thus, all the ERA-Interim and


NCEP-FNL data at 06:00 UTC were selected uniformly for information extraction
and anomaly recognition. The daily averages and the maximum and minimum values
based on the data from the ground-based stations were analyzed subsequently. In
addition, we used the $5^{th}$ and $95^{th}$ percentile box plots of $AOD_{532nm}$ each day to
effectively express the variations in the daily averages and maximum and minimum
values as a result of the differences in the daily data records [Che et al., 2014].
Second, the spatial distributions of the SML1, STL1, TMP2m, and PWATclm data
were analyzed. Considering the complex influences and possible uncertainties with
regard to seasons, terrain, weather, and latitude, we obtained the differential images of
the changed parameters ($\Delta P$) by subtracting the 2009 daily value from the means from
multiple years. The result reflected a normal background, i.e.,
$$\Delta P_t = P_t - \mu_t = P_t - \frac{1}{n}\sum_{i=1}^{n} P_i \qquad (1)$$

where $P_t$ is the daily value of a parameter in 2009 and $\mu_t$ is the corresponding daily
mean estimated over the years 2000-2008. The $\Delta P_t$ images on the same day in 2006
were applied for comparison, and the $P_i$ for 2006 was adjusted to the means of 2000–
2008, except 2006.
**2.2 Spatio-temporal features of hydrothermal parameters**
*2.2.1 SML1, STL1, PWATclm, and TMP2m from assimilation datasets*
In the coversphere, the soil is an important layer for the transmission of mass and
energy from the lithosphere to the atmosphere. The hydrologic conditions and thermal
properties of soil could be disturbed in the seismogenic process. Our intuitive analysis
showed that the variation curve of the SML1 parameter of the epicenter pixel
appeared to decrease from March to April in 2009 and 2006 (Fig. 2.1a). However, five
anomalies exceeded $\mu+1.5\sigma$ before the 2009 L'Aquila main shock, with the maximum
anomaly occurring on March 5. In the context of the gradual seasonal increase of
STL1, its anomalous variation became obvious on March 30, with the value being the
maximum for that month (Fig. 2.1b). Although the variation amplitudes of SML1 on
March 29 and 31 were less than those of the former two peaks, these dates were
quasi-synchronous with STL1 (Fig. 2.1b). Hence, the water content and temperature





in the soil significantly changed at end of March 2009. Comparing the PWATclm
behavior in 2009 with its relative stable fluctuation in 2006, which acts as the normal
background, the PWATclm parameter exhibited evident peaks on March 29, 30, and
31; the highest value reached 27.8 kg/m$^2$, which significantly exceeded $\mu+2\sigma$ (Fig.
2.1c). PWATclm represents the total water vapor content of the atmosphere column; in
this work, this parameter indicated that the moisture budgets on the surface and
atmosphere layer were disturbed by something abnormal. Air temperature is a direct
parameter related to the thermal variation in the coversphere. In our study, we also
found continuous anomalous peaks of TMP2m from March 29 to April 1, 2009. The
values in this time window exceeded $\mu+2\sigma$, except on April 1 (Fig. 2.1d). Considering
the reported anomalies in Table 1, we propose that the quasi-synchronous period
characterized by multiple parameter anomalies preceding the L'Aquila EQ is likely
the time window from March 29 to April 1, 2009. The details of the abnormal
deviation of the parameters during this time window are shown in Table 2.
We mapped the image series of each $\Delta P_t$ as the difference between the daily value and
the historical mean ($\mu$) to investigate the spatio-temporal evolution of the investigated
parameters. Figure 2.2a shows that the area with an abnormal increment of ΔSML1
was located in the L'Aquila basin on March 29 and 31, 2009 and that the local
ΔSML1 reached 19.5 K to 21 K in the epicenter grid. By contrast, the spatial pattern
in 2006 was characterized as normal with clear homogeneity for the land in central
Italy (Fig. 2.2b). This result implied that the moisture on the upper soil layer of the
seismogenic zone abruptly increased before the main shock. Although significantly
anomalous ΔSML1 occurred in north Italy, such anomaly was assumed to be unrelated
to the L'Aquila EQ because of its large area and remote distance. Different from that
of ΔSML1, the spatial anomalous field of ΔSTL1 initiated on March 29 and appeared
distinguishably northwest to the epicenter of the main shock on March 30, 2009 (Fig.
2.3a), especially along the southern segment of the *Olevano–Antrodoco* thrust (Fig. 1).
This abnormal pattern did not appear in 2006 (Fig. 2.3b). According to the local
meteorological data (Fig. 2.4), the particular spatio-temporal evolution of ΔSML1 and
ΔSTL1 did not result from precipitation. As changes in soil water stimulate thermal



change, a short delay in the change in soil temperature relative to soil moisture is
possible. In this work, we revealed a one-day delay between the increases in soil
temperature and soil moisture.
In the case of an abnormal variation in temperature and moisture in the soil layer,
hydrothermal conversion becomes increasingly significant on the surface and in the
atmosphere because of the wide, open space. Compared with PWATclm in almost all
the Italian territories and surrounding seas during the silent period, ΔPWATclm
showed a sudden increase on March 29, 2009; it then quickly dropped to a relatively
normal level on March 31, similar to the case in 2006 (Fig. 2.5). Although the
abnormal area of ΔPWATclm covered the entire Italy, a weaker abnormal area
appeared in the L'Aquila basin on March 29-31 and extended to the southeast (Fig.
2.5a), where it equaled ΔSML1 on March 29-31 (Fig. 2.3a). We considered the
possibility of the regional anomalous signal related to the seismogenic process being
masked by an intensive air–sea interaction in a large area on those days. Obviously,
both the spatial anomalies of ΔSML1 and ΔPWATclm were not controlled by
topographic conditions. Particularly, the normal spatial pattern of ΔTMP2m in central
Italy on March 28 to April 1, 2006 was slightly higher than that over the sea and
notably lower than that at the northern border of the Italian territory (Fig. 2.6b).
However, an anomalous spatial distribution of ΔTMP2m occurred on March 28 and
30, 2009, mainly in the intermountain area northwest of the main shock epicenter (Fig.
2.6a). The anomalies of the four investigated parameters were distributed mainly in
the L'Aquila basin or in the intermountain area northeast of the main shock epicenter
on March 29–31. Thus, we inferred that the regional topography (Apennine range and
L'Aquila basin) and tectonics (*Olevano–Antrodoco* and *Gran Sasso* thrusts) in central
Italy could have induced the spatial correlations of these anomalies.

### *2.2.2 Air temperature and AOD from ground-based stations*

To investigate possible thermal fluctuations in situ and to support the potential
coupling effects of such fluctuations on the ground surface, we collected air
temperature and AOD data from ground-based stations. Figure 2.6 shows that the




daily averages and the maximum and minimum values of air temperature at the
L'Aquila weather station reached their peaks on March 29 and 30, 2009 (Fig. 2.7).
Figure 2.8 shows the AOD variations that fluctuated in three time windows of the
abrupt AOD increase on March 16, 30, and April 3–6, 2009. The dates in which the
anomalous values of air temperature and AOD were observed were consistent with
those for SML1, STL1, PWATclm, and TMP2m. In particular, the general AOD values
were less than 0.3 (Fig. 2.8a); however, the maximum $AOD_{532nm}$ reached 0.37, 0.3,
and 0.46 in the three time windows, whereas the rest of the $AOD_{532nm}$ varied around
0.07–0.26, which is the same as those on clear days (Fig. 2.8b). Although the Roma
station of AERONET is far from the epicenter and the increase in AOD was weak, the
observed AOD data somehow served as the reference value for L'Aquila. The
secondary organic aerosol (SOA) in the atmosphere is generated from the
photochemical reaction of gas phase precursors, such as sulfur ($SO_2$) and nitrogen
($NO_2$) volatiles, as well as ozone ($O_3$) [Janson et al., 2001; Rickard et al., 2010],
whereas the photochemical production of $O_3$ is a result of the photo-oxidation of
methane ($CH_4$) and carbon monoxide (CO) [Dentener et al., 2006; Crutzen, 1974].
The increased $CH_4$ degassing soon after the L'Aquila EQ [Voltattorni et al., 2012;
Quattrocchi et al., 2011] could be hints of $O_3$ precursors. Hence, the anomalous
increments of aerosol might have been caused by the formation of SOA particulates as
a result of the photochemical production of $O_3$ from degassed $CH_4$. In addition, the
low precipitation in March 2009 (Fig. 2.4) indicated that the weather condition during
this period was acceptable and that the anomalous PWATclm increment in the
epicenter pixel on March 29, 2009 was not caused by rainfall.
**2.3 Summary of the hydrothermal parameter analysis**
The following seismic anomalies were determined to be possible according to the
quasi-synchronism analysis of the abnormal changes of six hydrologic and thermal
parameters related to the coversphere and atmosphere and according to the spatial
evolution analysis of the images of the changed values. 1) The anomalies were
observer mainly in the L'Aquila basin southeast of the main shock epicenter (ΔSML1





and ΔPWATclm) or in the Apennines range northwest of the main shock epicenter
(ΔSTL1 and ΔTMP2m). 2) The spatial migration of the hydrologic and thermal
changes in the upper soil layer (ΔSML1 and ΔSTL1) could have indicated the
reformation and redistribution of mass and energy transmitted from the lithosphere to
the coversphere. 3) The spatial distribution of the increased air temperature near the
surface was consistent with that of the soil temperature. Hence, the thermal
transmission process was stable from the coversphere to the atmosphere and was
controlled by regional tectonics in central Italy. 4) Although the improvement in the
precipitable water content in the atmosphere on March 29, 2009 was masked by its
high values in the surrounding large area, the anomalous weak values in the L'Aquila
basin suggested that the water in gaseous or liquid state was influenced by the soil
structure (aquifers) and surface topography. Considering these findings, we propose
that the anomalies be interpreted as the geosphere coupling effects preceding the 2009
L'Aquila EQ.

### 3.  Seismic *b*-value

**3.1 Data and method**

The EQ catalog for computing the *b*-value in this work was obtained from INGV
(ISIDE: http://iside.rm.ingv.it). This catalog covers all of Italy and its surrounding
regions. We analyzed the seismic data covering the periods from April 16, 2005 to
December 19, 2012, during which 94,953 events were recorded. Considering data
quality and tectonic regimes related to the 2009 L'Aquila EQ, we excluded in the
analysis the events that occurred at a depth of over 40 km and limited the study area to
the region within the 80 km radius of the epicenter of the L'Aquila main shock. Figure
3.1 shows the cumulative number of the analyzed data as a function of time. For the
curve shows a usual behavior until the end of 2009, we preferred to limit the analysis
to November 2009. Hence, all the succeeding analysis refers to the periods of August
2005 to November 2009. Referring to the changes in the slope of the plot of the
cumulative number of events, we identified three-staged phases of different recording
qualities, with P1-1 and P1-2 denoting the conditions before the 2009 L'Aquila EQ





and P2 denoting the conditions after the 2009 L'Aquila EQ. The details are as follows.
Phase P1-1: 3,552 events from April 18, 2005 to August 15, 2007;
Phase P1-2: 2,742 events from August 16, 2007 to April 5, 2009;
Phase P2: 19,782 events from April 6, 2009 to November 30, 2009;
In seismology, the classical Gutenberg–Richter law [Gutenberg and Richter, 1944] is
introduced as follows:

$$LogN(M) = a - bM \qquad (2)$$

where $N$ is the number of EQs with magnitudes greater than or equal to $M$ in a given
region and in a time interval; $a$ and $b$ are constants that describe the productivity and
relative size distribution of the area of concern, respectively. The study of the $b$-value
has been widely performed [Mogi, 1962; Urbancic et al., 1992; Warren and Latham,
1970], and its variations have been found to be caused by regional stress, material
properties, and temperature gradient. Using the software package ZMAP [Wiemer,
2001], we computed the maximum-likelihood $b$-values with the following Eq.(3):

$$b = \frac{\log e}{\overline{M} - M_O + \frac{\Delta M}{2}} \qquad (3)$$

where $\overline{M}$ is the mean magnitude and $Mo$ is the minimal magnitude of the given
sample; $\Delta M$ is the uncertainty in magnitude estimation and is usually set to 0.1. The
sample was considered complete down to the minimal magnitude $Mc \leq Mo$, which
also referred to as the magnitude of completeness [Schorlemmer and Wiemer, 2004].
To detect the dynamic features of the $b$-values, we estimated the $b$-values with Eq. (3)
in moving (partly overlapping) time windows. Generally, the sampling window
contains 200 seismic events, 10% of which is the sliding/overlap window (i.e., 20
events), $b$-value actually is estimated from part samples before and after each time
node. To visualize the spatial distribution of the $b$-values, all events in the study area
were projected onto a coordinate plane with a gridded space of 0.1 °longitude by 0.1 °
latitude. At each grid node, we sampled all the events within a radius of 20 km and
determined their $b$-values if at least $N_{min} = 30$ events were available. Following the
work of De Santis et al. [2011], we also calculated the corresponding Shannon



entropy of the EQ related to the *b*-value, i.e.,
$$H(t) = k - \log b(t) \ (k \approx 0.072) \tag{4}$$
This entropic quantity allows the measurement of the level of disorder of the seismic
system and the missing information or uncertainty because it is universally considered
a fundamental macroscopic physical quantity that describes the properties of complex
geosystemic evolutions, such as that of the seismogenic system in the lithosphere [De
Santis et al., 2011].
**3.2 Spatio-temporal features**
To compare the results of De Santis et al. [2011], we also reduced the catalog by Mc =
1.4 for the time series analysis of the *b*-values from phase P1-2 to phase P2.
Following the initial stable phase in 2008, the *b*-value drastically decreased as the
main shock approaching. Figure 3.2 shows that the curve drops to the lowest point of
*b* = 0.747 about March 27, 2009, i.e., ~10 days before the main shock or a few days
before the occurrence of various thermal anomalies [Piroddi and Ranieri, 2012;
Piroddi et al., 2014]. Meanwhile, the entropy gradually increased to reach the peak
(almost 0.2, Fig. 3.2) during the same period after a long (almost) stable period and
then dropped one week before the main shock. Note that the exact time when the
peaks were reached (minimal *b*-value and maximal entropy) could not be detected
properly because ZMAP applies a moving sliding window containing 200 events for
computation. Hence, each curve was slightly affected by what was preceding and
what was following the given moment of estimation. Moreover, the *b*-value (or the
entropy) appeared to have moved rapidly to the minimum (or the maximum) on the
day of the main shock. This condition indicated that the regional stress was rapidly
released and that faults ruptured quickly close to the main shock. Both the *b*-value and
entropy were unstable after the main shock because of the aftershocks. Although we
used a moving window with ZMAP to calculate the *b*-value, and, in turn, the entropy,
the minimum value of *b*-value and the maximum value of the entropy just around the
time of the mainshock is real and not an artefact. Fig. 3.3 shows a smaller interval of
time where the entropy has been estimated in subsequent non-overlapping intervals of



seismic events each: it is clear from the observed estimates (triangles) the
beginning of the increase of the entropy well before the mainshock (when the entropy
exceeds two times the standard deviation, sigma, estimated over the whole interval),
with maximum at around the moment of it (when the entropy exceeds even ten times
the standard deviation). For a better visualization of the observed general behaviour of
the entropy, we also draw the gray curve that defines a reasonable smoothing of the
entropy values: 15-point FFT (Fast Fourier Transform) before the mainshock and
50-point FFT smoothing after the mainshock. The different kind of smoothing is
related to the different rate of seismicity before and after the mainshock.
Then we split the catalog into two subsets in terms of their magnitudes, which were
lower than the estimated completeness values, i.e., Mc = 1.2 and Mc = 1.0. The spatial
distributions of the $b$-values clearly differed in the two phases before the L'Aquila EQ
(Fig. 3.4b and d). In phase P1-1, the $b$ values in the L'Aquila basin and its
surroundings were about 1.0, which indicated a normal regional stress level because $b$
= 1.0 is a universal constant for EQs in general [Schorlemmer et al., 2005; Kagan,
1999]. The anomalous areas of high $b$-values ($b \geq 1.2$) were located in the south and
east of the impending L'Aquila hypocenter. By contrast, some external areas with low
$b$-values were not relevant to the seismic sequence because of existing rare
hypocenters (Fig. 3.4a). However, most of the relative high $b$-values in phase P1-1
changed to extremely low $b$-values ($b \leq 0.8$) in phase P1-2. In particular, a relatively
homogeneous strip of low $b$-values extended westward from the hypocenter and
crossed the southern segment of the *Olevano–Antrodoco* thrust. This effect indicated
the development of rock mass fracturing in the east-to-west direction, especially in the
south of the impending hypocenter. Coincidentally, this strip representing a high stress
level was consistent with the location of the strongest variation in soil temperature on
March 30 (Fig. 2.3a). Most of the other parts along the *Olevano–Antrodoco* and *Gran*
*Sasso* thrusts retained relatively high stress levels, which implied low seismicity. The
changed spatial patterns of the $b$-values from P1-1 to P1-2 indicated the adjustment of
the regional crust stress to a relatively high level in the seismogenic zone before the
L'Aquila EQ. These conditions clearly reflected the intensive seismicity and



significantly rapid accumulation of crustal stress occurring near the approaching
L'Aquila main shock hypocenter relative to other places. Figure 3.3f shows the spatial
distribution of the *b*-values after the L'Aquila EQ. Different from that happened
before the main shock, the low *b*-values occurred in the L'Aquila basin and its
surroundings because of the fault rupture and the subsequent aftershocks (Fig. 3.4e).
We also notice that the extremely low *b*-values (red area) covered the entire *Gran*
*Sasso* thrust and the footwall of the *Olevano–Antrodoco* thrust. This observation
indicated that the developed cracks and ruptured rocks, which resulted from the
normal faulting of the L'Aquila EQ, passed through the entire *Gran Sasso* thrust but
stopped at the footwall of the *Olevano–Antrodoco* thrust.
We also selected the geological section (section 1 in Fig. 3.5a) used by Piroddi et al.
[2014] to show the variations in the *b*-values with depth before the L'Aquila EQ.
Another section of equal length (section 2 in Fig. 3.5a), which was perpendicular to
section 1 crossing the epicenter, was analyzed to identify further the differences in the
stress distribution and rock failure between section 1 and 2. Events above depth = 20
km were sampled to calculate the *b*-values in a buffer of 20 km from the two section
lines (Fig. 3.5a). In Fig. 3.5b, the spatial distribution of the low *b*-value appeared
around the hypocenter and extended about 25 km to SWW of the hanging wall of the
*Paganica* fault along section 1. This distribution illustrated the stress accumulation at
a depth of 10 km, which is shown as a stripe in Fig. 3.4d. A relatively low *b*-value
zone was observed between the *Paganica* fault and the *Gran Sasso* thrust. In addition
to the area of the impending hypocenter, the spatial image of the *b*-value along section
2 confirmed that the low *b*-value zone was near the *Gran Sasso* thrust and about 20
km from NNW of the *Olevano-Antrodoco* thrust (Fig. 3.5b). According to this result,
the geo-zones of stress concentration and rock failure were related not only to the
normal seismogenic fault (*Paganica* fault) but also to the two large thrusts
(*Olevano-Antrodoco* and *Gran Sasso* thrusts) long before the L'Aquila EQ. The
lowest *b*-values centered on the hanging wall of the *Paganica* fault at depths of 5-15
km (Figs. 3.5b and c). As shown in the vertical imaging section, the low *b*-values
partly connected the *Paganica* fault to the *Gran Sasso* thrust. Moreover, the relations



between the *b*-values and the geological depth in the whole study area were mapped
to investigate the change in the stress environment of the deep earth at different
phases (Fig. 3.6). We observed a similar variation trend of the *b*-values spatially
related with depth before (phase P1-2) and after (phase P2) the main shock. The
general *b*-value curves at both phases initially increased from 20 km to about 12.5 km,
rapidly dropped to the minimum at 9.5 km, and finally increased to high values at 5
km, which is the lowest depth indicated in the hypocenter records. Hence, the regional
crust stress accumulated at a depth of about 9.5 km, whereas the stress dropped at the
deep and low crusts. The stress change was stable at a depth of more than 20 km in
the study area. Obvious curve reversals appeared twice at depths of 8–12.5 km before
the main shock (Fig.3.6a). Hence, heterogeneous litho-stratigraphic
properties affected rock failure and led to different stress states in the study area.
According to CROP 11 ("CROsta Profonda," literally "Deep Crust") studies on the
near-vertical seismic reflection profiles crossing central Italy, which were supported
by The CROP Project and were initiated in the mid-1980s with joint funding from the
National Research Council, AGIP Oil Company, and ENEL (National Electric
Company) [Di Luzio et al., 2009; Patacca et al., 2008; Tozer et al., 2002], the
anomalous curve reversals resulted from the litho-stratigraphic difference among the
Mesozoic *Gran Sasso–Genzana* unit, *Queglia* unit, *Morrone–Porrara* unit, and even
the western *Marsica–Meta* unit. These carbonate units mainly contain
shallow-platform dolomite and limestone, which were overlaid disconformably by
Miocene carbonate deposits and siliciclastic flysch deposits. In addition, the *Queglia*
unit and the deeper *Maiella* unit contain Messinian evaporite and marl. Thus, the
anomalous reversal of the *b*-value with depth could be the result of the unconformable
Mesozoic–Cenozoic contact; the mixed flysch, evaporate, and marl might have also
affected the counter-regulation of stress accumulation. Hence, we infer that $10 \pm 5$ km
was the main depth range of the seismic stress variation associated with the L'Aquila
EQ.



### 3.3 Summary of the seismic analysis

The time series analysis of the *b*-values in phases P1-2 and P2 shows that after late December 2008, the *b*-value (or the entropy) rapidly went to the minimum (or the maximum), specifically on March 27, 2009 (10 days before the main shock), and then wildly fluctuated closely before and after the main shock. The date of occurrence of the anomalously low *b*-value coincided with that of the reported thermal anomalies, which indicated the rapid release of crust stress and fracturing of rock mass and/or faults. Compared with that in phase P1-1, the image of the *b*-value in the latter phase P1-2 showed abnormally low *b*-values near the impending L'Aquila hypocenter, as well as a homogeneous strip of low *b*-values extending toward the east-to-east direction and crossing the southern segment of the *Olevano–Antrodoco* thrust. After the main shock, the anomalous zone of low *b*-values emerged in the L'Aquila basin and its surroundings because of rupturing and subsequent aftershocks. The 3D spatial variation of the *b*-value showed that the zone of low *b*-values obviously appeared around the hanging wall of the *Paganica* fault at a depth of 5–15 km and extended to 20 km SWW. Similar anomalies of low *b*-values closely related to two large thrusts were also observed in NNW of the impending hypocenter. In particular, anomalous reversals of the *b*-values occurred twice at a depth of 8-12.5 km before the main shock, thus implying that unstable stress state did relate to heterogeneous litho-stratigraphic properties. The revealed spatial pattern of the *b*-values indicated that the space evolution characteristics of the stress accumulation prior to and immediately after the L'Aquila EQ reflect the spatial correlations among the L'Aquila EQ and seismic faults in the central Apennines.

### 4. Discussions

As mentioned above, SML1, STL1, TMP2m, PWATclm, *b*-value, and even AOD have quasi-synchronous time windows of anomalies. Obviously, this characteristic is not a simple coincidence, but its geophysical mechanism necessitates further analysis. Here, we attempt to provide a possible explanation in view of geosphere coupling.

### 4.1 Lithosphere: deep fluid and stress




The central Apennines are affected by a NE-SW striking extension and uplift. This
extension was responsible for the formation of intra-mountain basins, i.e., L'Aquila
basin, bounded by the *Gran Sasso* and *Mt. d'Ocre* ranges. During the L'Aquila
seismic sequence, the seismic events were focused on the upper parts of the crust with
a depth < 15 km; three main faults were activated by dip-slip movements in response
to the NE-SW extension [Di Luccio et al., 2010]. The ultimate cause of an EQ is
undoubtedly the crust stress exceeding the elastic limits of faults or rock mass. Crust
stress is indeed affected by particular geo-environmental conditions, including faults,
cracks, rock, and fluids, inside the lithosphere. Some studies based on the
measurements of the ratio of compressional velocity to shear velocity and of seismic
anisotropy have provided evidence that high-pressure fluid contributed to the
rupturing of the 2009 L'Aquila EQ [Di Luccio et al., 2010; Terakawa et al., 2010;
Lucente et al., 2010]. The contribution of fluids to the L'Aquila seismic sequence
evolution was independently confirmed by Cianchini et al. [2012] through magnetic
measurements from the L'Aquila geomagnetic observatory. As a result of the eastward
migration of the compressive front since the early Miocene, the back-arc extension
affected the Apennines chain, which was previously controlled by compressive
tectonics [Di Luccio et al., 2010]. Normal faults formed the L'Aquila basin and
affected the Apennines chain in the Pleistocene period [Doglioni, 1995]; moreover,
several works have increasingly implicated fluids and their movement in the
generation of the L'Aquila EQ [Di Luccio et al., 2010; Terakawa et al., 2010; Lucente
et al., 2010]. Both the eastward compressive and NE-SW extensive stresses could
have contributed to the deep fluid migration to the potential epicenter area.
Subsequently, seismogenic faults became weak as a result of the high pressure of pore
fluid and consequently reduced the stress level needed to break the rocks [Hubbert
and Rubey, 1959]. In particular, a proposed scenario suggested that the *Paganica* fault
plane initially acted as a barrier to fluid flow [Lucente et al., 2010]; hence, the fluid
pressures at both sides of the fault were unbalanced. The foreshock sequence,
especially the Ml 4.0 foreshock on March 30, broke the barrier, thereby allowing
fluids to migrate across the fault and change the Vp/Vs ratios [Lucente et al., 2010].





The migrating fluids would have dilated the rock mass of the hanging wall and
facilitated fault movement, leading to EQ nucleation. The images of the *b*-values in
phase P1-2 (Fig. 3.4d) and along the two orthogonal sections (Fig. 3.5) clearly show
the spatial distribution of the intensive stress accumulation and rock failure
development around the impending hypocenter and the large thrust at a depth of $10 \pm$
5 km in the crust, which correspond to fluid migration and high pore pressure,
respectively.
At this point, we clarify basic issues on fluids. First, we discuss the composition of
fluids and their sources. The Apennines located at the plate boundary are
characterized by high heat flow and large-scale vertical expulsion, volcanoes, gas
vents, mud pools, geysers, and thermal springs, which are typical surface features of
fluid expulsion [Chiodini et al., 2004; Chiodini et al., 2011; Minissale et al., 2004].
Two of the largest aquifers covering the Abruzzi region are the *Velino* and *Gran Sasso*
aquifers (Fig. 4.1b), which consist of Meso-Cenozoic carbonate formations (limestone
and dolomite) of the Latium–Abruzzi platform and of platform-to-basin transitional
domains [Chiodini et al., 2011]. For the fluid solution, the rich groundwater breeds an
ideal geo-zone for gas–water–rock reactions. Fluids with $CO_2$-rich gases are known to
be involved in the EQ preparation process [Di Luccio et al., 2010; Terakawa et al.,
2010; Lucente et al., 2010; Chiodini et al., 2011]. Both the numerous $CO_2$-rich gas
emissions mainly from the Tyrrhenian region and the large amounts of deeply derived
$CO_2$ dissolved by the groundwater of the aquifers of the Apennines have been
supported by geochemical and isotopic data [Chiodini et al., 2000, 2004, 2011;
Minissale et al., 2004]. The melting of the crust sediments of the subducted Adriatic–
Ionian slab is a regional $CO_2$ source, and the subsequent upwelling of the mantle and
the carbonate rich melts would have induced the massive degassing of $CO_2$ on the
Earth's surface [Frezzotti et al., 2009]. Thus, we conclude that the large quantities of
$CO_2$ gas in the two aquifers not only comprise a large portion of the dissolved
inorganic carbon derived from the Tyrrhenian mantle wedge and/or Adriatic
subducted slab in the deep Earth but also involve the progressive decarbonation of
minerals of the carbonate formations in the shallow crust.



Second, we explain how fluids migrate. On the one hand, Chiodini et al. [2011]
compared the geochemical composition of Abruzzi gas and that of 40 large gas
emission sites located in central Italy and found that the former becomes
progressively rich in radiogenic elements ($^4$He and $^{40}$Ar) and $N_2$ from the volcanic
complexes in the west to the Apennines in the east, thereby indicating the increasing
residence time of the gas in the crust moving from west to east. On the other hand,
Minissale et al. [2004] performed a systematic analysis of published geochemical and
isotopic data (together with new data) from the Apennines, including thermal and cold
springs, gas vents (mostly $CO_2$), and active and fossil travertine deposits, and found
that meteoric water precipitating in the high eastern Apennine ranges mixes with
ascending eastward magmatic, metamorphic, and geothermal fluids in the highly
permeable Mesozoic limestone.

**4.2 From lithosphere to coversphere: Earth degassing**

Before the main shock, $CO_2$-rich gases from different sources were involved in the
crustal circulation of fluids, and the mixed fluids could have been injected into the
regional groundwater system (i.e., *Velino* and *Gran Sasso* aquifers) and moved up to
the surface. Hence, the influx of $CO_2$-rich gases can increase pore pressure and flow
rate. During the foreshock sequence, the development of fractures and cracks of rock
mass would have facilitated the flow of fluids outside the aquifers in the shallow crust,
which is bordered by the *Olevano–Antrodoco* and *Gran Sasso* thrusts. Meanwhile,
electronic charge carriers of crustal rocks in the form of peroxy defects known as
p-holes [Freund, 2011] could have been activated when the rock was stressed.
Overpressured fluid could further reduce the friction of fault planes and reactivate
faults. As a result of the widespread aquifers and the high permeability of carbonate
formations, underground fluids with $CO_2$-rich gases easily migrated upward to the
coversphere under the accelerated stress condition. The rising of shallow underground
fluid alters soil physical properties (i.e., soil moisture and temperature) and thereby
affects different components of surface energy balance. A gas geochemical monitoring
conducted in a natural vent close to the L'Aquila basin observed anomalous $CO_2$ gas





flow variations in March and April 2009 [Bonfanti et al., 2012]. The intensive $CO_2$
degassing from ground measurements confirms the emission of deeply originating
gaseous fluids to the coversphere. The increase in greenhouse gas emission (i.e., $CO_2$,
$CH_4$), is an important mechanism of pre-EQ thermal anomalies. In addition, as radon
gas might cause air ionization and variations in humidity and latent heat exchange, the
anomalous Rn emanation before the L'Aquila EQ was recorded [Pulinets et al. 2010].
Soil gas surveys [Voltattorni et al., 2012; Quattrocchi et al., 2011] revealed $CO_2$ and
certain amounts of $CH_4$ and Rn as released gas phases. Hence, we propose that the
degassing of $CO_2$, $CH_4$ and Rn from the lithosphere to the coversphere before the
main shock could have resulted in the complex lithosphere–coversphere coupling
effect, which finally increased near-surface temperature and generated heavy TIR
emissions.

**4.3 From coversphere to atmosphere: air ionization**

As a transition layer from the lithosphere to the atmosphere, the coversphere affects
the flow and exchanges of mass and energy from the deep crust to the surface. As
revealed by the ESA global land cover data produced from the Medium Resolution
Imaging Spectrometer sensor aboard the Envisat satellite, the thermal anomalous zone
based on the Night Thermal Gradient (NTG) algorithm (Fig. 4.1a) was mainly
covered by high vegetation, i.e., broadleaved deciduous forest, with strong water
retention and developed root traits. Generally, high vegetation coverage represents
high moisture in deep soil and improves the active characteristics of surface soil,
including organic matter contents, which promote fluid concentration and movement
through preferential flow and root absorption [Chai et al., 2008; Millikin et al., 1999]
Hence, we propose that high vegetation in central Italy facilitated the upward
migration of $CO_2$-rich fluids inside the coversphere before the 2009 L'Aquila EQ. We
also suggest that this upward migration of $CO_2$-rich fluids generated heavy thermal
radiations because surface temperature rise results from possible greenhouse effects
together with latent heat release stimulated by the decay of radon and/or the activation
of P-holes.



Air ionization is a fundamental factor of energy balance in the lower atmosphere.
When underground gases are released on the surface, the air composition of the lower
atmosphere must change. The leaked $CO_2$ and $CH_4$ gases on the surface can serve as
radon carriers, and α-particles emitted by a certain amount of decayed radon can
further motivate the air ionization process [Pulinets and Ouzounov, 2011]. In addition,
the activated p-hole outflow leads to air ionization at the ground–air interface [Freund,
2011]. Hence, both radon emanation and P-hole activation processes could have
contributed to the air ionization and resulting ion hydration before the 2009 L'Aquila
EQ. The direct results of ion hydration are humidity change and latent heat release. In
turn, increased latent heat changes the content of water vapor. In this work, the local
greenhouse effect and latent heat release jointly resulted in the increase in air
temperature, and TIR anomalies (i.e., NTG) were observed by satellite sensors before
the 2009 L'Aquila EQ. Ion hydration in the air requires particulate matter as water
condensation nucleus after air ionization; hence, aerosol particle injection (AOD
increase) is theoretically necessary [Qin et al., 2014b].
Although rock failure developed mainly in the hypocenter area and related to normal
faulting, the *b*-value features of the thrusts in the wing of the *Paganica* fault indicate
that NTG thermal anomalies are indeed related to compressive stress. Some key
matters, such as $CO_2$, $CH_4$, and radon, can be enriched at a shallow depth and
transported to the surface along the two seismic faults to finally cause regional
thermal anomalies. The hypocenter area is bounded by two intersecting thrusts, with
the *Olevano–Antrodoco* thrust being the main one. The experimental TIR observations
on the fracturing of loaded intersected faults revealed the close relationship between
the changed TIR radiation and the geometrical structure of intersected faults, with
abnormal TIR spots usually occurring along the main fault [Wu et al., 2004, 2006]. In
addition, two separate zones of surface NTG anomalies (Fig. 4.1a) could have
different modes from deeper thermal sources.
Therefore, a particular LCA coupling mode is proposed to interpret the
comprehensive geophysical mechanisms of multi-parameter anomalies associated
with the 2009 L'Aquila EQ. Before the main shock, the deep $CO_2$-rich fluids changed



the geo-environment in the lithosphere, including the geophysical properties of rock
mass, the chemical composition of groundwater, and fault activity. Thus, the resulting
intensive crust stress varied in the specific area, particularly in the southern segment
of the *Olevano–Antrodoco* thrust. Forced by the resulting intensive stress and driven
by high-pressure fluids, abnormal gas matters (including $CO_2$, $CH_4$, and Rn) and heat
energy moved up to the coversphere and altered the water content and temperature in
the soil layer (i.e., SML1 and STL1). Furthermore, soil and vegetation facilitated the
upward migration of $CO_2$-rich fluids to the atmosphere. In general, a chain of LCA
coupling effects related to the L'Aquila EQ occurred as 1) the upwelling of
underground fluids increased the soil temperature (STL1) and SML1; 2) the decay of
radon and the activation of P-holes led to air ionization; 3) the triggering of air
ionization and subsequent ion hydration were promoted by aerosol particle injection;
4) a series variation occurred in water and heat, including a drop in atmospheric
relative humidity, latent heat release, and change in water vapor (i.e., PWATclm); 5)
air temperature increased (i.e., TMP2m); and 6) TIR anomalies (i.e., NTG) were
observed from the satellite sensors.

## 5.  Conclusions

The anomalies of hydrothermal parameters in the coversphere and atmosphere before
the 2009 L'Aquila EQ appeared in significant quasi-synchronous time windows on
March 29-31, 2009 (three days). The spatial patterns of these anomalies were
controlled by the seismogenic tectonics and local topography. The temperature
variation of the soil and the near-surface atmosphere, which was mainly distributed in
the intermountain northwest of the main shock epicenter, indicated that the thermal
anomalies were geo-related to the large thrusts outside the rupturing zone. Moreover,
the zones of the most intensive soil and air temperature anomalies were consistent
with that of NTG from the satellite and with the increased *b*-value in phase P1-2. The
results related to the hydrographic and thermal anomalies in the coversphere and
atmosphere compensate for the deficiency in current interpretations on the LCA
coupling of the 2009 L'Aquila EQ. The supplemental temporal analysis of air



temperature and AOD further proved the dates of thermal anomalies and supported
the coversphere-atmosphere coupling effects.
As a parameter of stress meter, the *b*-value should be applied to EQ anomaly
recognition and the analysis of geosphere coupling effects to logically and spatially
link multiple observations on the coversphere and atmosphere with that on the
lithosphere. In this study, we deduced from the dynamic variation of the *b*-values that
the regional stress had started to rapidly accumulate in late December 2008 and soon
entered the nucleation stage. The end of March, 2009 was possibly a critical time
node of stress transition. The 3D variation features of the *b*-values revealed that the
regional crust stress accumulated to a relatively high level from phase P1-1 to phase
P2-2 in the hypocenter area before the main shock. The *b*-values notably decreased
after the main shock because of the aftershock sequence. Furthermore, the relation
between the *b*-values and the hypocenter depth indicated that the shallow crust with a
depth of less than 10 km was the main geo-layer characterized by a high stress level,
especially near the *Paganica* fault and the southern segment of the *Olevano–*
*Antrodoco* thrust. The depth of $10 \pm 5$ km was considered as the main depth range of
the crustal stress transition related to the 2009 L'Aquila EQ.
Regional/local tectonics, lithology, hydrogeology, geochemistry, and land cover have
great influence and/or control over the generation and spatio-temporal evolution of
multiple anomalies before a tectonic EQ. $CO_2$-rich underground fluids played a vital
role in the coupling processes from the lithosphere to the coversphere in the 2009
L'Aquila EQ because their characteristics benefitted the migration of mass and energy
from the lithosphere to the coversphere. Hence, to clearly understand the phenomena
and mechanisms of anomalous signals related to tectonic EQs, we need to pay close
attention to local geological, hydrogeological, and geographical environments. The
coversphere is a key part of geospheres and has a major effect on the production and
transmission of seismic signals as well as anomalies. Knowledge of the coversphere is
extremely important in studying the mechanism and physical process of LCA or LCAI
coupling before tectonic EQs. Moreover, certain particular matters in the deep Earth,
such as deep-originated fluid, including water and gases, should be investigated to



analyze and understand the observed pre-EQ anomalies.
**Acknowledgements**
This work is supported by the National Basic Research Program of China (973
Program) (Grant No.2011CB707102) of the China Ministry of Science and the
Technology, and some parts of this work has been performed under the auspices of the
SAFE (Swarm for Earthquake study) ESA-funded Project. The hydrothermal
parameters were obtained freely by ERA-Interim (http://apps.ecmwf.int/datasets/data)
and NCEP-FNL (http://rda.ucar.edu/datasets). The AOD data were obtained freely
from NASA AORNET (http://aeronet.gsfc.nasa.gov/). The *b*-value data were
generated by Italy seismic catalog and obtained by INGV (ISIDE:
http://iside.rm.ingv.it). The seismic and weather data about L'Aquila have been
downloaded from two open-access (upon free registration) websites: The seismic
data from seismic catalog ISIDe (http://iside.rm.ingv.it/) maintained by the Istituto
Nazionale di Geofisica e Vulcanologia (INGV), Italy, while the wheather data have
been taken from http://cetemps.aquila.infn.it/ maintained by CETEMPS, Italy.

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

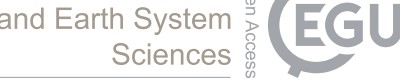


**Table 1** Reported multiple parameter anomalies associated with the Mw 6.3 2009 L'Aquila EQ

| *Parameters* | *Date of alternative anomalies* | *Geospheres* | *Reference* |
|---|---|---|---|
| Acoustic Emission | from 4th to 5th March | Lithosphere | Gregori et al., 2010 |
| Seismicity rate | from 27th March to 6th April | Lithosphere | Papadopoulos et al., 2010 |
| *b*-value | 27th March | Lithosphere | Papadopoulos et al., 2010 |
| Entropy of *b*-value | from 31st March to 6th April | Lithosphere | De Santis et al., 2011 |
| LF radio wave | from 31st March to 1st April | Lithosphere | Biagi et al., 2009 |
| ULF magnetic | from 29th March to 3rd April | Lithosphere | Eftaxias et al., 2009 |
| VLF electric | started on 1st April | Lithosphere | Rozhnoi et al., 2009 |
| $CO_2$ flow-rate | started on 31st March | Coversphere | Bonfanti et al., 2012 |
| Radon | started on 30th March | Coversphere | Pulinets et al., 2010 |
| Uranium groundwater | started at beginning of March | Coversphere | Plastino et al., 2009 |
| Land surface temperature | started on 29th March | Coversphere | Piroddi and Ranieri, 2012 |
| Thermal infrared radiation | from 30th March to 1st April | Coversphere/Atmosphere | Lisi et al., 2010 |
| Thermal infrared radiation | from 30th to 31st March | Coversphere/Atmosphere | Genzano et al., 2009 |
| F2-layer critical frequency | 16th March, 5th April | Ionosphere | Tsolis and Xenos, 2010 |
| Total electron content | 2nd April, 4th April | Ionosphere | Akhoondzadeh et al., 2010 |



















**Table 2** Hydrothermal parameter anomalies from March 29 to April 1, 2009 (quasi-synchronous
period)

| *Parameters* | *Alternative anomaly date* | *Abnormal deviation* | | *Spatial anomaly* |
|---|---|---|---|---|
| | | *> μ+1.5σ* | *> μ+2σ* | |
| SML1 (m³/m³) | March 29 | 0.007 | | Strongly concentrated in L'Aquila basin and geo-related to *Olevano-Antrodoco* and *Gran Sasso* thrusts to the north. |
| | March 31 | 0.003 | | Concentrated in L'Aquila basin and geo-related to *Olevano-Antrodoco* and *Gran Sasso* thrusts to the north, but it was smaller and weaker than that on March 29. |
| STL1 (K) | March 30 | 1.13 | | Strongly concentrated on the east of the mainshock with EW trending (crossing the southern part of *Olevano-Antrodoco* thrust) and extended to the northwest of the central Italy. |
| PWATclm (kg/m²) | March 29 | | 8.97 | Strongly covered the entire land and sea of the central and southern Italy and weakly concentrated on the east part of L'Aquila basin (the south of *Gran Sasso* thrust). |
| | March 30 | | 0.03 | Unapparent |
| | March 31 | | 0.11 | Unapparent |
| TMP2m (K) | March 29 | | 0.53 | Strongly and largely distributed in the northwest of the central Italy (to the west of *Olevano-Antrodoco* thrust). |
| | March 30 | | 1.05 | Unapparent |
| | March 31 | | 1.28 | Distributed in the northwest of the central Italy (to the west of *Olevano-Antrodoco* thrust), but was smaller and weaker than that on March 29. |
| | April 1 | 0.571 | | Unapparent |






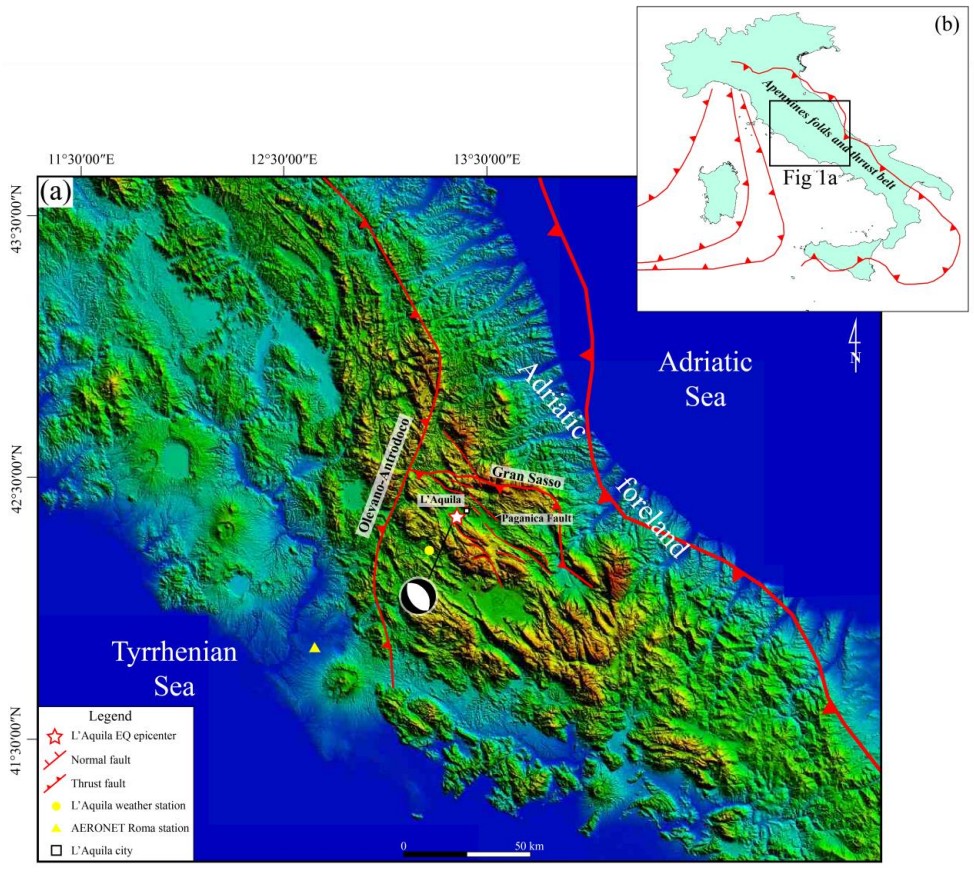

**Fig. 1** Simplified tectonic and topographic map of central Italy. (a) DEM from the Shuttle Radar Topography Mission (SRTM) data showing the epicenter of the 2009 L'Aquila EQ (white star) along with its focal mechanism solution (FMS). The FMS was obtained from the US Geological Survey's National Earthquake Information Center (Di Luccio et al., 2010; Piroddi et al. 2014). (b) The main thrusts in Italy, with the black solid line box showing the geographical location of Fig. 1a (Benoit et al. 2011).





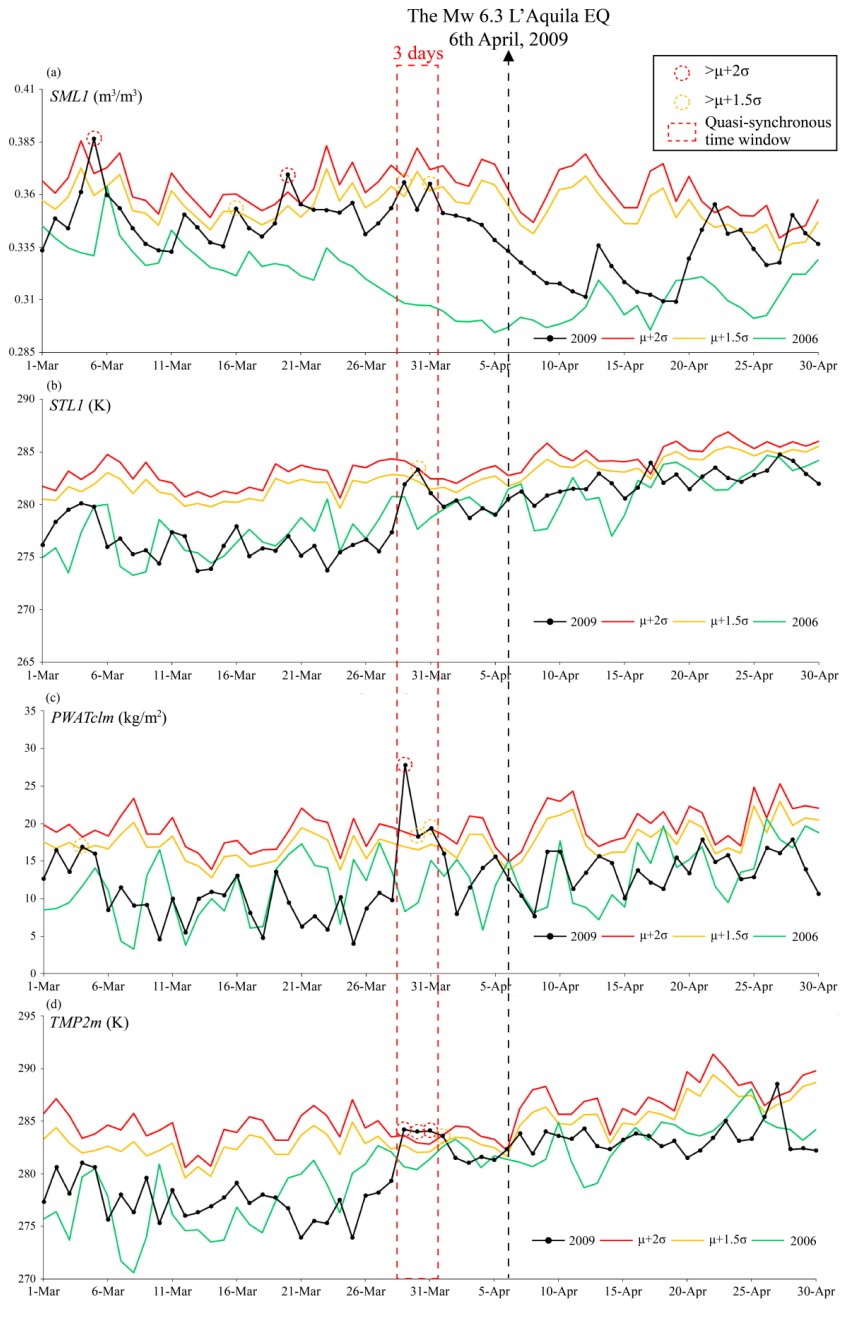


**Fig 2.1** Time series of four hydrothermal parameters, *SML1* (a)*, STL1* (b)*, PWATclm* (c) *and*
*TMP2m* (d), on the epicenter pixel from March to April 2009, and its comparison with historical
data over the same period. The red and orange lines show the value of ($\mu+2\sigma$) and ($\mu+1.5\sigma$),
respectively; the green and black lines show the value in 2006 (as a normal background) and 2009,
respectively.





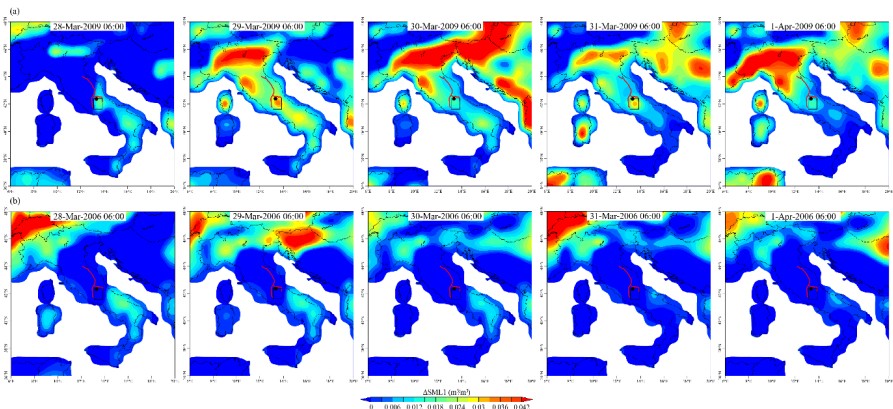

**Fig. 2.2 S**patial distributions of ΔSML1 at 06:00 UTC from March 28 to April 1, 2009 (a) and 2006 (b), respectively. The black spot indicates the epicenter of the main shock, the black rectangular box indicates the epicenter pixel, and the red line indicates the related main fault system.

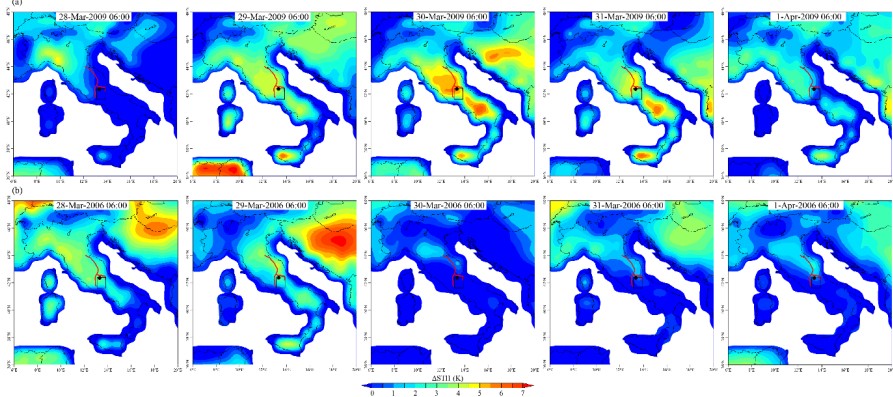

**Fig. 2.3** Spatial distributions of ΔSTL1 at 06:00 UTC from March 28 to April 1, 2009 (a) and 2006 (b), respectively. The black spot indicates the epicenter of the main shock, the black rectangular box indicates the epicenter pixel, and the red line indicates the related main fault system.

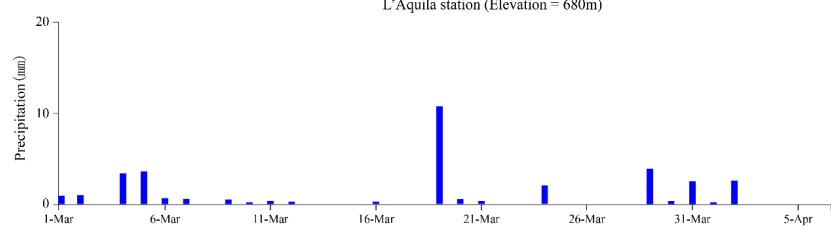

**Fig. 2.4** Daily precipitation at L'Aquila station from March 1 to April 5, 2009.



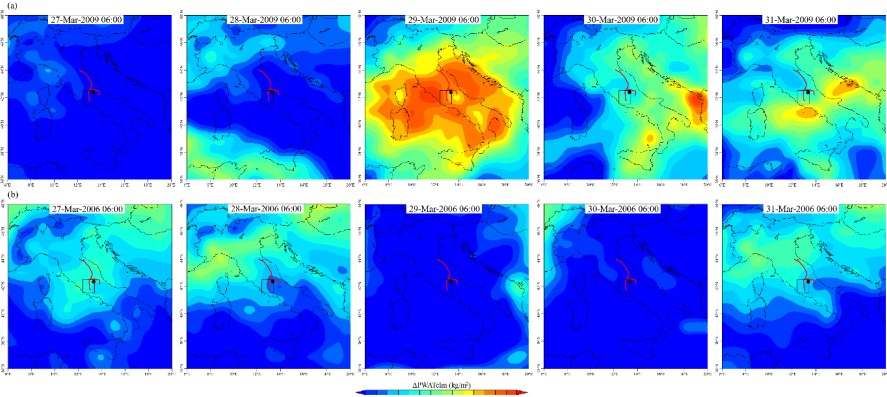


**Fig. 2.5** Spatial distributions of ΔPWATclm at 06:00 UTC from March 28 to April 1, 2009 (a) and 2006 (b), respectively. The black spot indicates the epicenter of the main shock, the black rectangular box indicates the epicenter pixel, and the red line indicates the related main fault system.


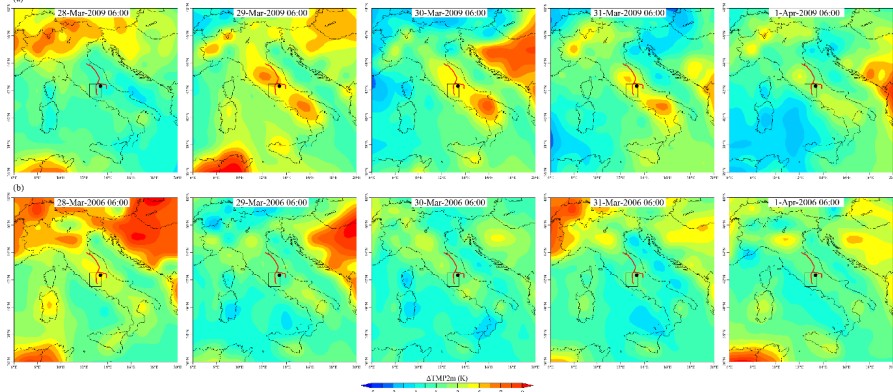


**Fig. 2.6** Spatial distribution of ΔTMP2m at 06:00 UTC from March 28 to April 1, 2009 (a) and 2006 (b), respectively. The black spot indicates the epicenter of the main shock, the black rectangular box indicates the epicenter pixel, and the red line indicates the related main fault system.






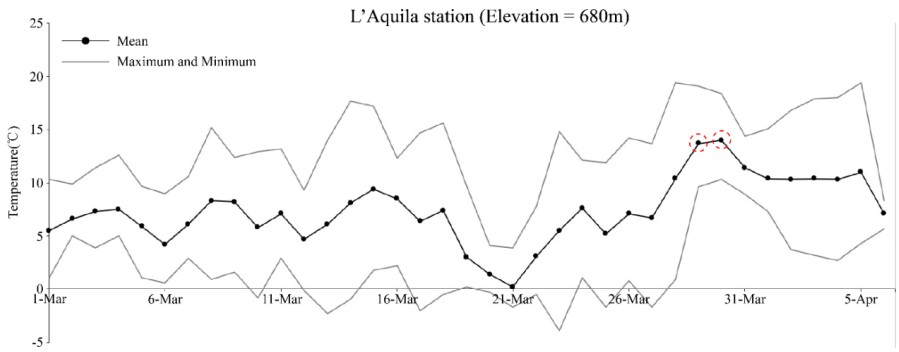


**Fig. 2.7** Daily average and maximum and minimum values of air temperature at the L'Aquila
station from March 1 to April 5, 2009.

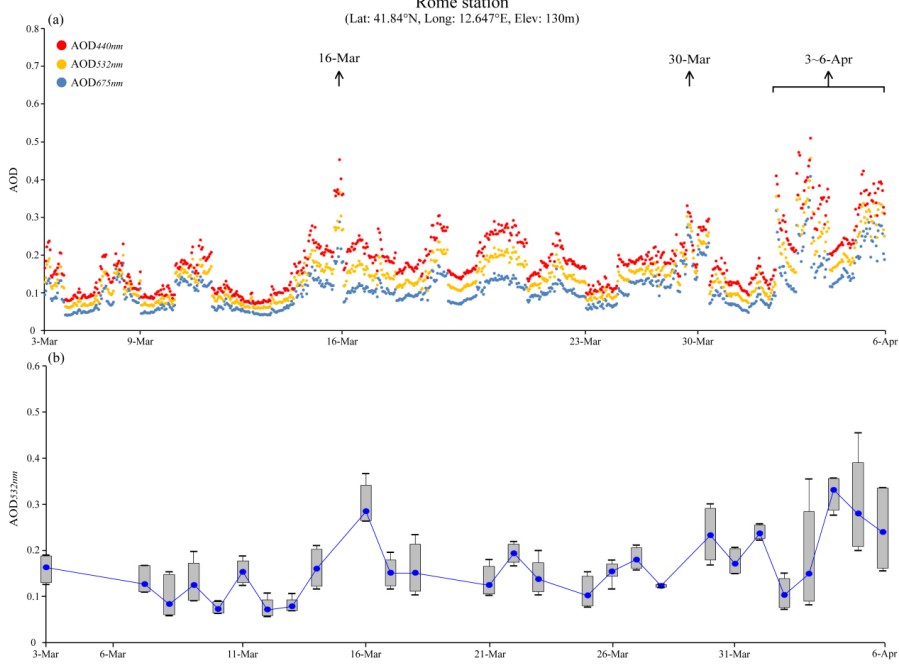


**Fig. 2.8** Time series of AOD at the Roma station of AERONET from March 3 to April 6, 2009. (a)
AOD at 440, 532, and 675 nm; (b) daily average and maximum and minimum values of $AOD_{532nm}$,
as well as the 5th and 95th percentile box plots.





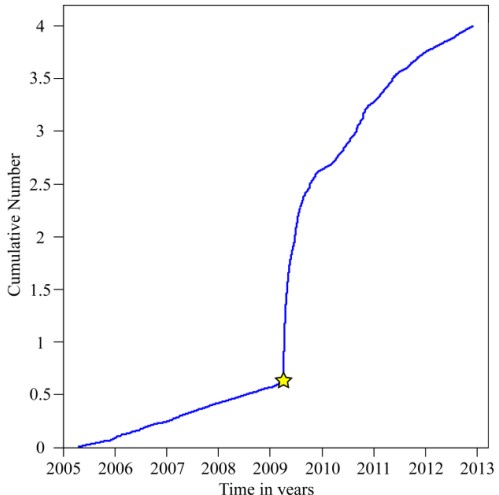

**Fig. 3.1** Cumulative number of analyzed catalog as a function of time (the yellow star shows the main shock of the 2009 L'Aquila EQ).

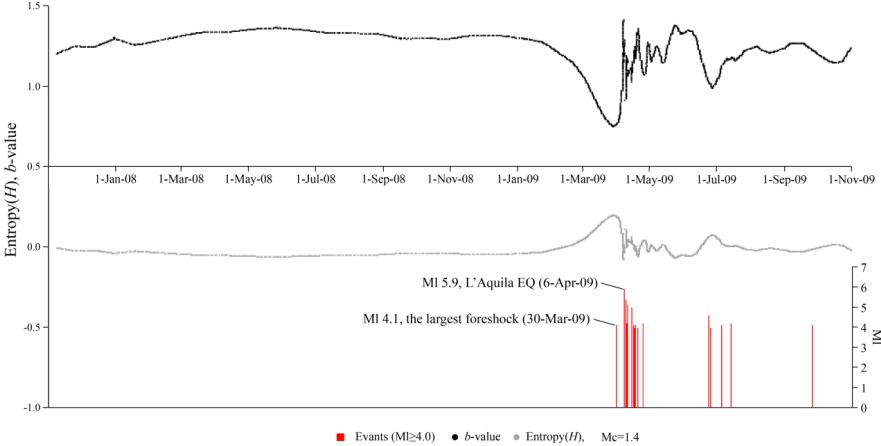

**Fig. 3.2** Time series of *b*-value (above plot), Shannon entropy (*H*; intermediate plot), and seismic events (Ml ≥ 4.0; bottom plot) during phases P1-2 and P2.





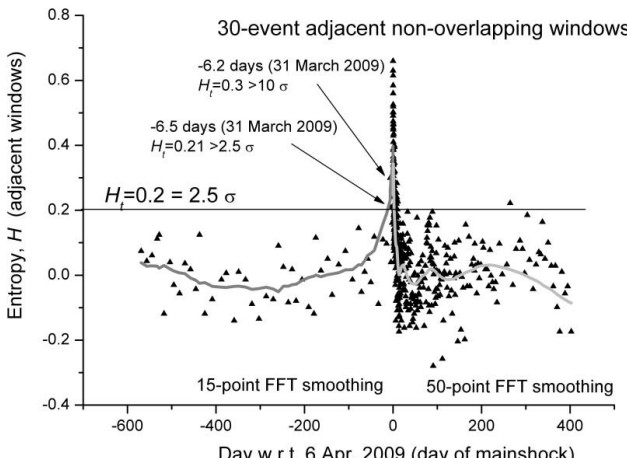

1082

**Fig. 3.3** Shannon entropy for L'Aquila seismic sequence from around 1.5 year before the mainshock to around 1 year after, calculated for a circular area of 80 km around the mainshock epicenter. The gray curve defines a reasonable smoothing of the entropy values: 15-point FFT before the mainshock and 50-point FFT smoothing after the mainshock. Sigma is the standard deviation estimated over the whole interval.



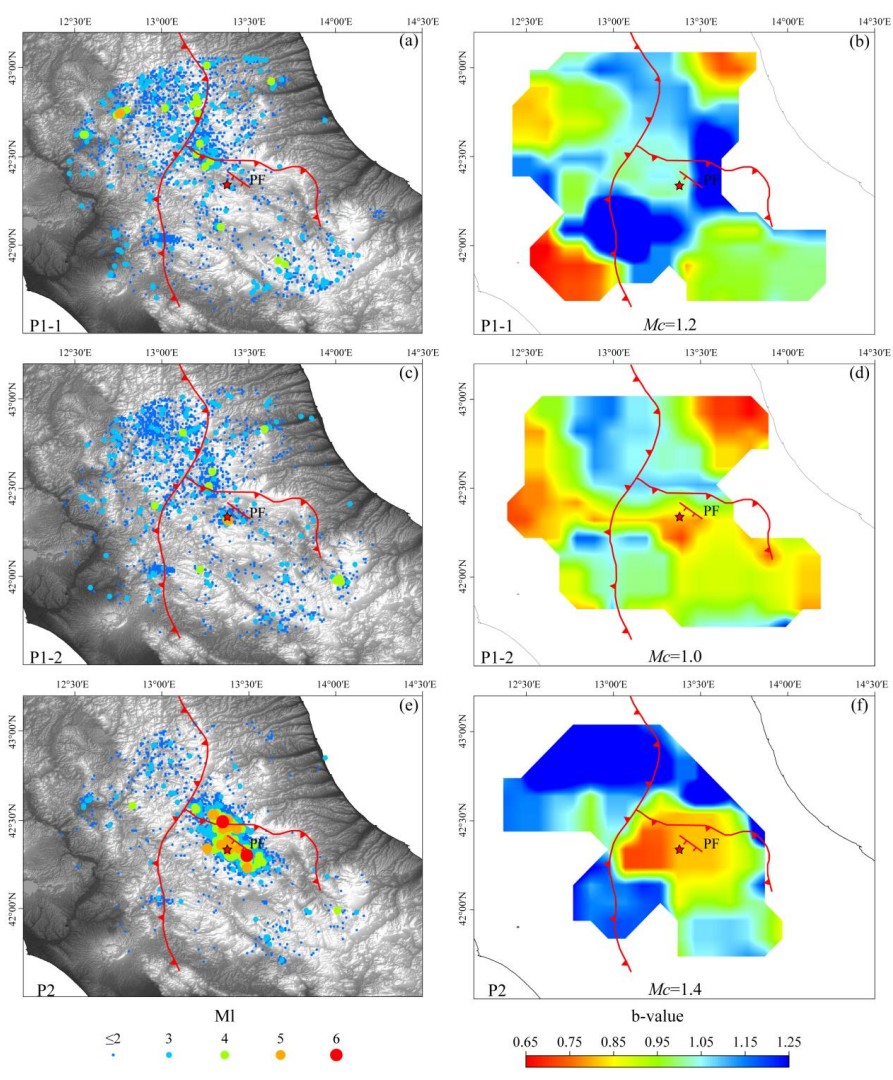

**Fig. 3.4** Spatial distributions of epicenters (a, c, e) and *b*-value (b, d, f) before and after the main shock of the L'Aquila EQ at three-staged phases (the red star and the red lines represent the main shock and main fault system, respectively).


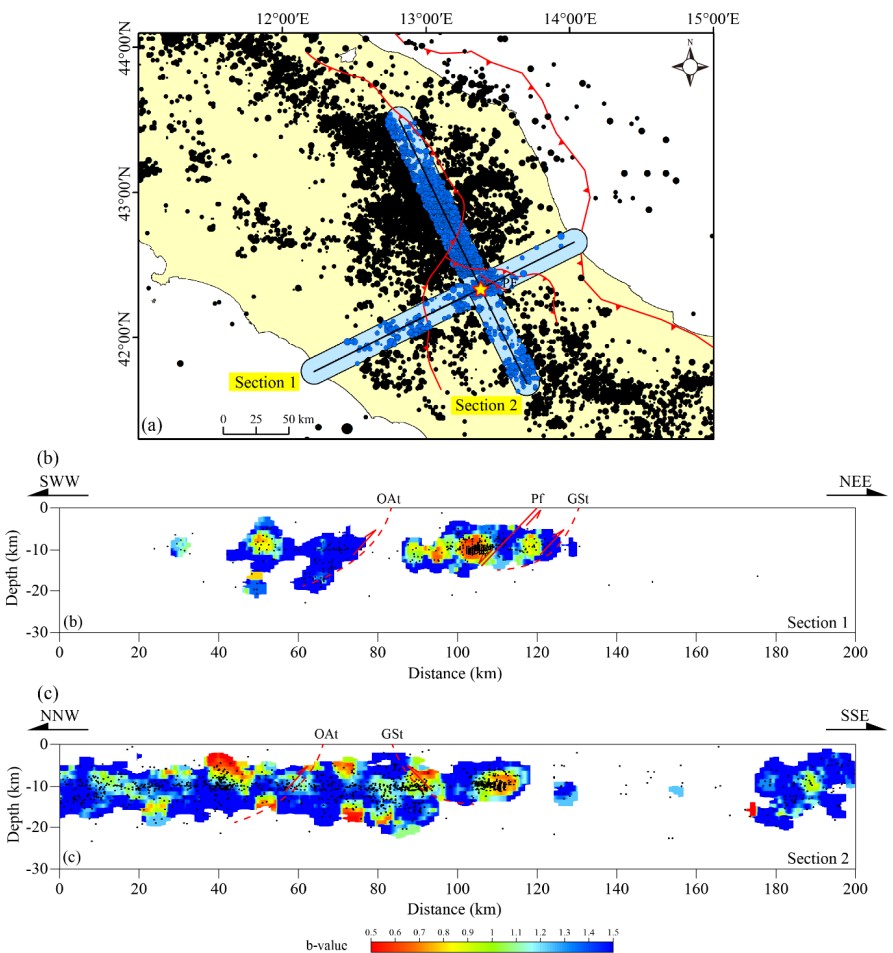

**Fig. 3.5** Spatial distribution of epicenters/hypocenters and *b*-values from P1-1 to P1-2. (a) The outcrops of seismic faults and thrust, as well as all the epicenters of the foreshocks and the main shock; (b) *b*-values along section 1 crossing the main shock epicenter and seismic faults and thrust (the black dots represent hypocenters); (c) *b*-values along section 2 crossing the main shock epicenter and seismic faults and thrust (the black dots represent hypocenters). OAt: *Olevano–Antrodoco* thrust, GSt: *Gran Sasso* thrust, Pf: *Paganica* fault.



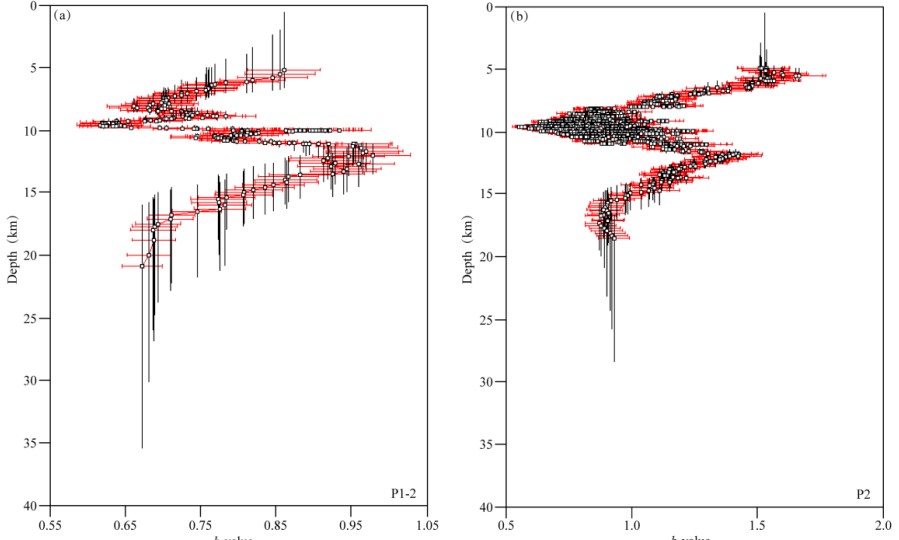

**Fig. 3.6** The relation between the *b*-values and hypocenter depths in phase P1-2 (a) and phase P2
(b). The dots indicate the average *b*-values related to depth, the horizontal bars indicate the
uncertainty in the *b*-values, and the vertical bars indicate the depth range of the sampled
hypocenters.

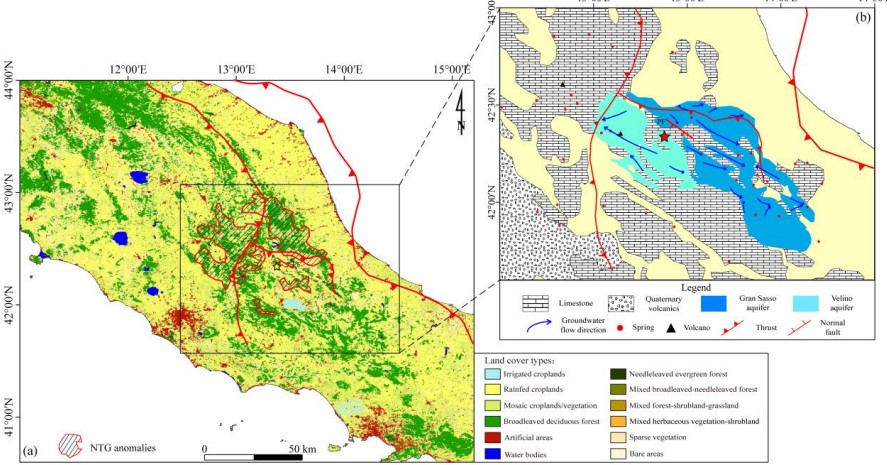

**Fig. 4.1** An integrated representation of the geographical (coversphere) and geological
(lithosphere) environments associated with the 2009 L'Aquila EQ. (a) Zones of NTG anomalies
from LST data overlapped by land covers (Piroddi and Ranieri, 2012); (b) the spatial distribution
of tectonic faults, geological rocks, hydrogeological aquifers, and groundwater flows in the
epicenter area and its surroundings (Chiodini et al., 2012).





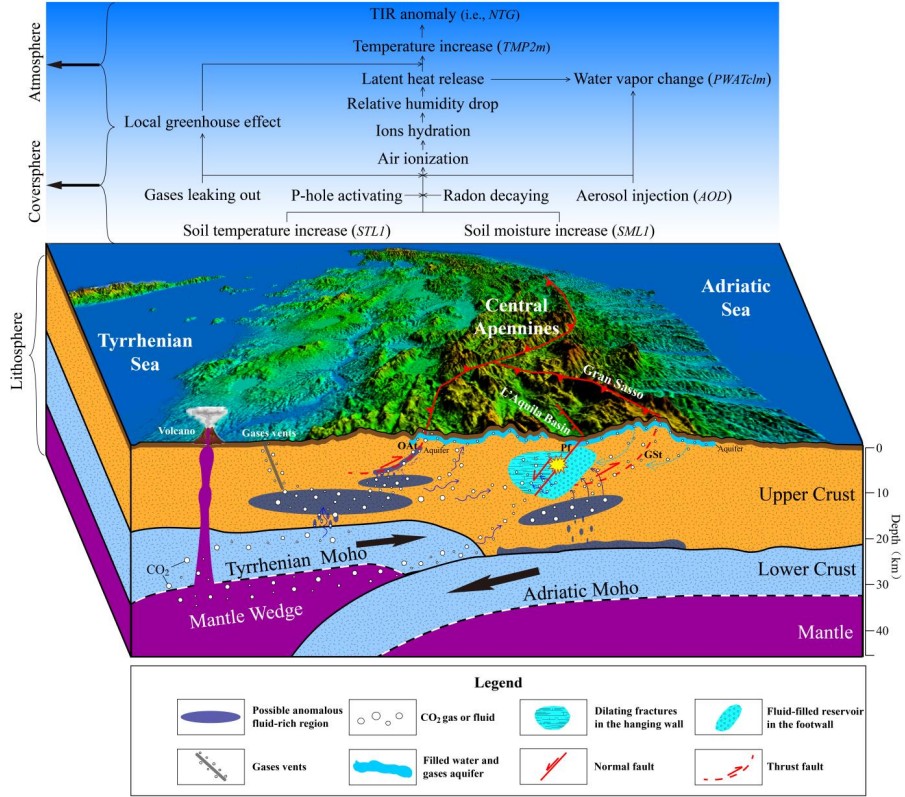


**Fig. 4.2** Mechanism of hydrothermal anomalies and conceptual mode of LCA coupling associated
with the 2009 Mw 6.3 L'Aquila EQ in Italy (referring to Chiarabba et al., 2010; Chiodini et al.,
2004; Di Luccio et al., 2010; Lucente et al., 2010; Terakawa et al., 2010). OAt: *Olevano–*
*Antrodoco* thrust, GSt: *Gran Sasso* thrust, Pf: *Paganica* fault.