# Peer review of "Geosphere Coupling and Hydrothermal Anomalies before the 2009 Mw 6.3"

_Natural Hazards and Earth System Sciences, 2015_

## Short Comment (SC1) · 4 Apr 2016

Wu et al. (cited as WU16 below) investigate the possible occurrence of precursors of the 6 April 2009 L'Aquila earthquake in atmospheric and soil parameters (cited as ASPR below) by means of retrospective analyses of assimilation datasets (soil moisture, soil temperature, near-surface air temperature, and precipitable water) and in ground-based observations of temperature and atmospheric aerosol, as well as in seismic data. The authors show preearthquake changes that they interpret as possible precursors of the earthquake.

Reports of earthquake precursors have a serious social responsibility because they motivate the idea that in the future we will able to predict earthquakes. Thus, the identification of actual and reproducible precursors is the key point for evaluating the potentiality of an earthquake prediction method.

Even if the subject of WU16 is very timely, as there are still debates on the existence of earthquake precursors, on their detection, as well as on possible methods of earthquake prediction, the manuscript shows many weaknesses that cast serious doubts on the seismogenic origin of the reported preearthquake changes. Below there are some remarks on how the authors introduce the topic of earthquake precursors and on their analysis of ASPR. This brief comment does not address the analysis of seismic data.

1) The introduction section of WU16 is very one sided in favor of earthquake precursors and does not correctly introduce the state-of-the-art in the search for precursors. The authors in support of their findings quote papers claiming the observation of precursors of the 2009 L'Aquila earthquake. Thus, in WU16 the authors report that a *large number of the precursory anomalies of the 2009 L'Aquila EQ were reported.* In fact, what they claim is not correct. All the changes (in any geophysical parameter) identified to precede the earthquake of L'Aquila are to be considered only alleged precursors because actual evidence of the relationship between these changes and the earthquake has never been provided. Moreover, the authors seem to show a poor knowledge of the recent literature in the topic of earthquake precursors, as well as of the studies on the precursors of the L'Aquila earthquake. Below there are some examples that show how the Introduction section of WU16 is biased.

– The authors, while mentioning Biagi et al. (2009) as a paper showing an actual precursory anomaly in LF radio signals, are unaware that Biagi and his co-authors (see Biagi et al., 2010) refute their previous findings reported in Biagi et al. (2009).

– WU16 quote Eftaxias et al. (2009) as report of actual precursors of the 6 April earthquake. Eftaxias et al. (2009) show electromagnetic anomalies at the Greek station of Zante (800 km away from L'Aquila) that they claim to be precursors. Note that analysis of data from L'Aquila area did not identify any electromagnetic and co-seismic signature that may be actually recognized as seismogenic, and alleged precursory signatures claimed to be identified in these local data have been shown to be actually unrelated to the earthquake (see Biagi, 2009, 2010; Masci, 2012; Masci and De Luca, 2013; Masci and Di Persio, 2012; Villante et al., 2010).

– WU16, while mentioning papers proposing possible physical mechanisms for the generation of preearthquake electromagnetic signals, are unaware of recent laboratory experiments on fluid-saturated rock samples that do not support the hypothesis that electromagnetic signals may be generated during the slow stress accumulation that may occur prior to earthquakes (see Dahlgren et al., 2014).

2) One of the main shortcomings of WU16 is the identification of precursory signatures in ASPR (see Section 2). In general, the authors do not provide a rigorous qualitative definition of what constitutes an anomaly, nor do they show if the alleged anomalies appear only before the earthquake, or whether they appear frequently, more or less at random.

- Page 6, row 182: Why preearthquake anomalies are more remarkable at 06:00 UTC?

- Assimilation datasets

  The method used by WU16 for identifying pre-earthquake anomalies cannot be considered a valid method to find earthquake precursors. For example, in Figure 2.1 there is no physical reason that $(\mu+2\sigma)$ and $(\mu+1.5\sigma)$ represents the "normal background" above which seismogenic anomalies may be isolated.

  Still, the existence of a quasi-temporal synchronism is not very significant in order to claim a possible seismogenic origin of the reported anomalies because, as can be seen in Fig. 2.1, there is usually a correspondence between the changes in the ASPR parameters. STL1, e.g., shows a positive correlation with TMP2 during all the period shown in the figure.

  The authors show just over one month of data before the L'Aquila earthquake. Instead, they should demonstrate that actual seismogenic anomalies appear only before the earthquake, ruling out that these anomalies appeared frequently during the previous years.

- Ground-based datasets

  Once again the authors show just one month of data.

  The ground based datasets shown in WU16 are not indicative of the climatic conditions of the L'Aquila area.

  The air temperature station is close two lakes and is away from the fault that generated the earthquake (see the figure below)

[Figure]

The aerosol optical depth is obtained using data from a station very far from L'Aquila area. This station is in the highly urbanized area of the city of Rome close to the Ciampino airport (see the figure below).

[Figure]

3) What probably led the authors to look for precursors in a wide area is the paper by Dobrovolsky et al. (1979) where the authors report a theoretical formula for calculating the alleged preparation zone of the earthquake, a zone where physical phenomena should lead to the subsequent shock. Thus, this formula is usually used to support the observation of precursors away from the epicentral area. However, the theoretical formula of Dobrovolsky seems to be not supported by experimental evidence (see Masci and Thomas, 2014, 2015a, 2015b). Furthermore, if we accept that earthquake precursors may be observed in the area estimated by the Dobrovolsky's formula, this raises some doubts regarding the usefulness of precursors for developing short-term prediction capabilities of earthquakes. A prediction is a deterministic statement that a future earthquake of magnitude M will occur in a particular geographic region and in given period of time. For the Mw9.0 Tohoku-Oki, Japan, earthquake of 11 March 2011 the Dobrovolsky's formula estimates a preparation zone having a radius of 7413 km. Note that this represents approximatively one thirds of the Earth's surface. A precursor observed within this very wide area would not have been useful for predicting the Tohoku-Oki earthquake.

In summary, WU16 it is yet another paper that attempts to find earthquake precursors in geophysical parameters. Unfortunately, the authors do not provide any evidence that the identified pre-earthquake changes in ASPR are actually anomalous, and more importantly that their origin is actually seismogenic.

References:

- Biagi et al. (2009): doi:10.5194/nhess-9-1551-2009.
- Biagi et al. (2010): doi:10.5194/nhess-10-215-2010.
- Dahlgren et al. (2014): doi:10.1785/0120140144.
- Dobrovolsky et al. (1979): doi:10.1007/BF00876083.
- Eftaxias et al. (2009): doi:10.5194/nhess-9-1953-2009.
- Masci (2012): doi:10.5194/nhess-12-1717-2012.
- Masci and De Luca (2013): doi:10.5194/nhess-13-1313-2013.
- Masci and Di Persio (2012): doi:10.1016/j.tecto.2012.01.008.
- Masci and Thomas (2014):doi:10.1002/2014JA019896.
- Masci and Thomas (2015a): doi:10.1002/2015JA021336.
- Masci and Thomas (2015b): doi:10.1002/2015RS005734.
- Villante et al. (2010): doi:10.5194/nhess-10-203-2010.

---

## Referee Comment (RC1) · Anonymous Referee #1 · 14 Apr 2016

1. Interpret "LCA" and "b-value" in abstract. Do not use abbreviation "EQ" both in abstract and in main body. 2. Line 98 – Term "coversphere" looks very new, I prefer old "Earth's surface". "Entities must not be multiplied beyond necessity". From the other hand may be authors will give the definition of the term "coversphere". 3. Line 132 – It seems, air temperature and aerosol do not belong to hydrothermal parameters. Here and further authors implicate hydrogeological term "hydrothermal" with meteorological parameters on the Earth's surface related with water content and temperature. May be it is acceptable for meteorological audience, not for the NHESS. 4. Line 152 – ERA-Interim – describe please. 5. Line 154 - 512° etc. ??? 6. Line 177 – "alternative anomaly" - probably "probable". 7. Line 194 – formula looks bad. 8. Fig. 2.1 – do not use such abbreviation in axis. 9. Fig. 2.2-2.6 - do not use abbreviation such as STL in figure description. 10. I propose to reduce speculation about CO2, CH4, Rn

emission –Chapter 4.1 and further. Or, please, show some gas data and relation with meteorological parameters. 11. I also propose to reduce some figures in range 3.1 - 3.5. 12. Thermal anomalies and last figure 4.1 looks weak. May be author will use this publication, for example, N. Pergola, C. Aliano, I. Coviello, C. Filizzola, N. Genzano, T. Lacava, M. Lisi, G. Mazzeo, and V. Tramutoli. Using RST approach and EOS-MODIS radiances for monitoring seismically active regions: a study on the 6 April 2009 Abruzzo earthquake Nat. Hazards Earth Syst. Sci., 10, 239-249, doi:10.5194/nhess-10-239-2010, 2010

---

## Author Comment (AC1) · 24 Apr 2016

General repliesïijŽ We fully agree with Dr. F. Masci that the reports of earthquake precursors have a serious social responsibility, and that the identification of actual and reproducible precursors is the key point for evaluating the potentiality of an earthquake prediction method. The dynamic geosystem is very complex and extremely uncertain, which makes earthquake be difficult to be predicted. However, the integrated observations (such as Global Earth Observation System of Systems, GEOSS) is providing us more and more data of multiple parameters on planet Earth, among which some are embed with hints on seismogenic meanings and shocking precursors. We are stepping into an era of big data endowing geoscientists in geophysics, seismology, remote sensing, etc., with unprecedented opportunity to explore the possibility of earthquake prediction by way of mining potential EQ precursors from multiple observations and various data sets. Although there are still debates on the existence of earthquake precursors, it's worthy of going forward and seeking for seismicity-related anomalies at the first step. Who can say no before doing our best? In this paper, we investigate the hydrothermal anomalies before the 2009 Mw 6.3 L'Aquila earthquake by means of retrospective analyses of assimilation datasets (soil moisture, soil temperature, near-surface air temperature, and precipitable water). After comparing them to historical data of preceding nine year and referring to ground-based observations of temperature and atmospheric aerosol, as well as to seismic data (b-value), we interpret the anomalies as possible precursors of the main shock of 2009 Mw 6.3 L'Aquila earthquake. We are not alleging the reported hydrothermal anomalies be definite EQ precursory, but showing a good example how to link objectively multiple parameters for seeking potential EQ precursors.

Point-to-point repliesïijŽ

1) We affirm something true, i.e. "a large number of precursory anomalies of the 2009 L'Aquila EQ were reported", and we mention 15 references, as an incomplete list of references. In fact, our paper is not about discussing all previous work but analyzes some multi-parameter observations taken in and around L'Aquila epicentral area to detect unexpected changes with respect to the normal behaviors. - We thank the referee about the presumed anomaly in LF radio signals. We removed Biagi et al. 2009 from the text of paper and the list in Table 1. - We agree with the referee that the analysis by Eftaxias et al. (2009) is based on a rather distant EM station from L'Aquila, but we think is worth maintaining because this kind of analysis is quite original, with a rather different approach with respect to other papers. By the way, there is some works that is based on the geomagnetic data of L'Aquila Observatory that provides some interesting results (e.g. Cianchini et al. 2012), so we cannot completely confirm that those data contain significant precursory information but its extraction might depend on the data analysis. -We are aware of the paper by Dahlgren et al 2014. There was only one kind of igneous rock (Gabbro) tested in the experiment of Dahlgren et al, while the

geological body of L'Aquila is carbonate units (limestone and dolomite). Prof. Freund and his collaborators had comments (e.g. Scoville et al., Nat. Hazards Earth Syst. Sci., 15, 1873–1880, 2015) to the work of Dahlgren et al 2014. We have no intention to go into this dispute so we preferred to cite the most important papers on this subject.

2) The interactive processes behind the lithosphere-atmosphere-ionosphere coupling are very complex and we admit there is still great discussion on its real presence or not, and, in the former case, even about which coupling processes are in act. However, we actually provide a general and rigorous definition of "anomaly" in a statistical sense, in terms of a deviation with respect to a given threshold (least 1.5), quantitatively defined through a certain number of standard deviation, the latter estimated on the base of the previous 9 years (2000-2009) of data in the same period of the mainshock year for checking that anomalies appear frequently, more or less at random. - From previous works on satellite thermal data, the best observation times are usually those in the night or early morning, in order to keep avoid of disturbance from ground surface reflectance at day times, which could submerge any pre-EQ thermal anomaly. After careful check, we found that the detected thermal anomalies emerge more clearly at 06:00 UTC, so we preferred to consider and show the result for this time of the day. A possible reason could be that ground radiation at 06:00 UTC is relatively low in favor of detecting weak anomalies possibly related with the earthquake. - We admit that there is no physical reason to define the anomalous signal as that above 1.5 or, better, above times standard deviation However we find this definition quite operative, and general enough to be applied to all analyzed parameters, and confidence probabilities is 86.6% or 95.5% when 1.5 or 2 standard deviation from statistics concepts. It is also true that some trends in the ASPR parameters agree well each other, we notice that the most of times the single deviations are different. What is instead interesting is that around a week before all ASPR data overcome 1.5 or even 2 standard deviation, while the general climatic conditions were not such as to explain this anomalous behavior. The referee is misleading (perhaps by accident) when he says that we show one month of data before the EQ, because we actually compare the two month period around the EQ

in 2009 with the same period of the previous 9 years (2000-2008). In addition we also show the behavior of the 2006 when no significant seismicity occurred in the Abruzzi region. - We agree with the referee that the ground based datasets shown by us are not indicative of the climatic conditions of the L'Aquila area, only. Instead they are expected to be indicative of a larger area, and this explains why we used even weather station which are around 40 km from L'Aquila EQ epicenter (but within the Dobrovolsky area; see also below). By the way, as showing in the referee's Google map the air temperature station is not so close to two small lakes but in the mountain areas. The distances between the station and the two lakes are about 8 km and 20 km, respectively, which is not likely for the lakes to suppress the transient rapid temperature variations at station place (at meteorological scale). Of course, the lakes will have effect on air temperature at station place in climate scale. In addition, regarding the aerosol optical depth data with a station in the periphery of Rome (about 150 km to the L'Aquila epicenter), it is again surprising that the corresponding general signal was rather regular but with anomalies at times in agreement with the ASPR parameters. According to our previous study (Qin, K., Wu, L.X., Zheng, S., et al., Is there an abnormal enhancement of atmospheric aerosol before the 2008 Wenchuan earthquake? Advance in Space Research, 2014), the satellite observed abnormal AOD related to the 2008 Wenchuan EQ covers more than 200 km (see the figure below). Hence, the AOD anomaly could be regarded as potentially related with L'Aquila earthquake. Fig.D1 Abnormal satellite-based AOD pattern one week before the 2008 Wenchuan EQ (Qin et al., 2014b)

3) For Dobrovolsky et al. (1979) model and strain radius concept, this paper and the corresponding topic would require a dedicated discussion, but that we summarized for evident limits of space. That paper is theoretical and is based on a mechanical model of the volume of rock around the impending fault that will slip and cause the EQ. Under the tectonic stress that always acts at the lithospheric plates (reaching forces of the order of 1012-1013 N for each meter of tectonic margin), under reasonable hypotheses, there is a deformation over a larger presumed spherical volume, identified by the strain radius. This radius is the distance from the fault after which the strain is

comparable with terrestrial tides. Its value has only a practical importance to define the area where the strain effect during the preparation phase is not negligible and identified by instrument. It is instead clear that in general the closer to the fault the distance, the greater the expected effect. It is obvious that once there is a deformation, which is the driving effect, it could imply some other possible effect, such as thermal, EM, gas release etc., depending on the rocks, the geological and tectonic conditions, the fault styles and their synergic interactions and configuration. We admit that the larger the magnitude of the impeding EQ the larger the area interested in its preparation phase, so paradoxically the greater the size of the expected EQ and the wider the area interested, so the more difficult to identify the precise site of the impending EQ. But this does not mean that the research is not worth doing. Rather, our opinion is that the complexity of the phenomenon does not preclude to understand it in the next future, especially attempting to connect and explain the quasi- synchronism of the appearance of different anomalies, which are not only those ASPR but also seismic. The latter are consolidated and serve as optimal indication of the subsequent phases preceding the L'Aquila mainshock. Therefore we disagree firmly to what the referee affirms in his last sentence. Our paper is not another paper that attempts to find EQ precursors in geophysical parameters. This would be a too simple generalization that will reduce the importance not only of our work but even that of any kind of researches in this field. It is clearly a "one sided position". Of course it is not our intention to solve all still open questions. We instead attempt to connect temporally and spatially the behaviors of different parameters, seismicity included. We also provide a possible reasonable logic that relates these observations under the umbrella of space-and-time referring to lithosphere-coversphere-atmosphere-ionosphere coupling. We believe our efforts can be useful to the scientific community working on this difficult but interesting and important subject. We finally agree with one of the first sentences of the referee, i.e., that if we "investigate the possible occurrence of precursors of the 6 April 2009 L'Aquila earthquake in atmospheric and soil parameters and in ground-based observations of temperature and atmospheric aerosol, as well as in seismic data", and we improve our

understanding, the implications on the Society would be great.

Very Sorry for the delayed response because my travelling to Vienna to participate the EGU 2016 assembly.

Sincerely yours Lixin Wu (on behalf of all co-authors) April 24, 2016

Please also note the supplement to this comment: http://www.nat-hazards-earth-syst-sci-discuss.net/nhess-2015-346/nhess-2015-346-AC1-supplement.pdf

SYNTAM AOD 550nm 10x10km on 5 May 2008

**Fig. 1.** Abnormal satellite-based AOD pattern one week before the 2008 Wenchuan EQ

---

## Author Comment (AC2) · 24 Apr 2016

We thanks very much the Anonymous Referee #1 for his or her kind and detailed comments. Point-to-point reply: 1) We add interpretations of "LCA" and "b-value" in the abstract, i.e. add "lithosphere-coversphere-atmosphere" and "b-value (a seismicity parameter from Gutenberg–Richter law)", and replace all abbreviations "EQ" with "earthquake" in the paper.

2) We give a simple explanation of term "coversphere" in Line 55. The definition of "coversphere" could be referred to Wu et al., 2012. We use coversphere to emphasize the transferring of energy and mass from lithosphere to atmosphere through their interface media (coversphere), which has not been mentioned in Lithosphere-Atmosphere-Ionosphere coupling (LAIC) model. The coversphere is an integral representation of the three dimensional geometry, mass distribution and its spatial difference of the geo-entities (including soil and sand layers, surface water bodies, forests, and vegetation) between lithosphere and atmosphere. And, the word "coversphere" is in consistent with lithosphere, atmosphere, ionosphere. etc., in word structure for better understanding the sphere-shaped components of planet Earth. Hence, we use coversphere instead of Earth's surface.

3) We agree with you that air temperature and aerosol may be not belonging to hydrothermal parameters from hydrogeological term. Here we using "hydrothermal parameters" in a broad meaning which including the water phase change and energy exchange in process of lithosphere-coversphere-atmosphere coupling. By the way, the analyses of air temperature and aerosol in this paper were based on ground observations, and the using meteorological "air temperature" and aerosol is actually an additional manner to further illustrate the energy-mass exchanges among lithosphere, coversphere and atmosphere to validate possible mechanisms of pre-earthquake anomalies.

4) "ERA" is not an abbreviation but the capital of "era", we add some introduction of "ERA-Interim".

5) The "512° ...256°" description on the gridded data is wrong. Thanks again. We modify it to be "512 lines of longitude and 256 lines of latitude".

6) Thanks for this indication. We use "probable anomaly " instead of "alternative anomaly".

7) Thanks for the kind reminding. The Line 194 – formula is not bad, but the last sentence in the line 193 is wrong (some words was erased by mistake). We change "The result reflected a normal background" to be "The result reflected the current deviation of the parameter value referring to a normal background".

8) and 9) Thanks for the suggestions. We replace all of abbreviation with full words in

Fig2.1-2.6.

10) Actually, there are some reported/alleged gas anomalies about $CO_2$, $CH_4$, and Rn emission (Voltattorni et al., 2012; Quattrocchi et al., 2011; etc.) as introduced in Chapter 1 and Table 1. In Chapter 4, we use the alleged $CO_2$ anomaly (Frezzotti et al., 2009; Chiodini et al., 2000, 2004, 2011; Minissale et al., 2004) to speculate a possible LCA coupling mechanism with fluid between lithosphere blocks and fluid from lithosphere to coversphere and atmosphere; we use the alleged $CH_4$ anomaly (Chiodini et al., 2000, 2004 and 2011) to speculate a possible mechanism of AOD anomaly caused by emitted $CH_4$; and, we use the alleged Rn (Pulinets et al. 2010) to speculate a possible mechanism of hydrothermal anomaly due to emitted-Rn caused air ionization and variations in humidity and latent heat exchange. Although the gas data were not related with meteorological parameters and the speculations are based on reported anomalies and our limited knowledge, it is worthy of mention to assistant further understanding of LCA coupling related to L'Aquila earthquake, and it is of inspiration meanings to researches on other earthquake cases. So, we want to keep most of this speculations (in condition of paper-length permitted), but canceled some sentences (such as in line 639-642: Some key matters, such as $CO_2$, $CH_4$, and radon, can be enriched at a shallow depth and transported to the surface along the two seismic faults to finally cause regional thermal anomalies) . We expect your understanding and kind support.

11) Thanks for your nice suggestion. We remove Fig 3.1 from the paper in that it is relatively regular and similar work had been reported in De Santis, A. et al. (2011).

12) For the thermal anomalies related with earthquake are usually much weak, the anomalies in the figures (Fig 2.1, 2.3, 2.6, and 4.1) are not so "weak" (reached two times of standard deviation) as compared with normal background. The positive anomalies of $\triangle$STL1 and TMP2m are remarkable as represented in red-color in Figure 2.3 and 2.6, respectively. The representation of thermal anomalies in Figure 2.3 and 2.6 is in a wider region (the whole Italy) but not the very limited Abruzzi region, which may have miss led your reading. To make a spatial contrast among thermal anomalies,

tectonic faults, geological rocks, hydrogeological aquifers and landcovers, we use also "NTG" thermal anomaly (Piroddi and Ranieri, 2012) in Fig 4.1. In fact, the work of Pergola et.al 2010 is also very good, we had mentioned it in the introduction Chapter and added it in Table 1.

Very Sorry for the delayed response because my travelling to Vienna to participate the EGU 2016 assembly.

Sincerely yours Lixin Wu (on behalf of all co-authors) April 24, 2016

Please also note the supplement to this comment:
http://www.nat-hazards-earth-syst-sci-discuss.net/nhess-2015-346/nhess-2015-346-AC2-supplement.pdf

[Figure]

**Supplement:**

**Geosphere Coupling and Hydrothermal Anomalies before the 2009 Mw 6.3 L'Aquila Earthquake in Italy**

L.X. Wu[1,5*], S. Zheng[2], A. De Santis[3], K. Qin[1], R. Di Mauro[4], S.J. Liu[5] and M. L. Rainone[4]

*1 School of Environment Science and Spatial Informatics, China University of Mining and Technology, Xuzhou, China*

*2 Academy of Disaster Reduction & Emergency Management, Beijing Normal University, Beijing, China*

*3 Istituto Nazionale di Geofisica e Vulcanologia, Sezione Roma 2, Roma, Italy*

*4 Dipartimento di Ingegneria e Geologia, Chieti University, V. Vestini 31, 66013 Chieti Scalo, Italy*

*5 Northeast University, Shenyang, China*

*Corresponding author: Lixin Wu,

School of Environment Science and Spatial Informatics, China University of Mining and Technology, Xuzhou, China;

Email: awulixin@263.net, wlx@cumt.edu.cn

**Abstract:** The earthquake anomalies associated with the April 6, 2009 Mw 6.3 L'Aquila earthquake have been widely reported. Nevertheless, the reported anomalies have not been so far synergically analyzed to interpret or prove the potential lithosphere-coversphere-atmosphere coupling process. Previous studies on $b$-value (a seismicity parameter from Gutenberg–Richter law) 
[revised manuscript text omitted]

---

## Referee Comment (RC2) · S. Pulinets (Referee) · 5 May 2016

I consider that the paper of Wu et al. is the strong contribution and support to the plenty of papers devoted to the L'Aquila earthquake and especially to the precursors observed before this earthquake. In this regard the comment of F. Masci that there few publications on the L'Aquila case is absolutely wrong. The number of publications on the L'Aquila case will occupy several pages and I do not intend to make the bibliographic review and will stop only on few moments demonstrating that Masci is wrong.

1. Dobrovolsky earthquake preparation zone.

It is not only Dobrovolsky et al. conclusion but many authors from different countries and different specialties. The geochemists from France Toutain and Baubron in their paper:

Toutain J.-P., Baubron J.-C., Gas geochemistry and seismotectonics: a review, Tectonophysics, 304, 1-27, 1998

demonstrated that radon distribution before earthquake follows the Dobrovolsky zone, i.e. inside of the area rounded by Dobrovolsky radius.

Bowman, D.D., Ouillon, G., Sammis, C.G., Sornette, A., Sornette, D. An observation test of the critical earthquake concept, J. Geophys. Res. – 1998. – 103. – B10. – P. 24359–24372

discussing the critical concept in seismology confirm the Dobrobolsky zone approach. I cite from the text of the paper:

We now turn to a possible interpretation of figure 7, which suggests that the logarithm of the critical region radius scales directly with the magnitude of the final event in the sequence. A line with a slope of 1/2 gives an excellent fit to the data. Dobrovolsky et al. [1979] and Keilis-Borok and Kossobokov [1990] report a similar scaling log R = 0.43 M for the maximum distance between an earthquake and its precursors, based on a completely different procedure, namely the optimization of pattern recognition techniques [Gelfand et al, 1976].

As one can see, Keilis-Borok and Kossobokov also come to the same conclusion. All of the mentioned persons are the first rank scientists in seismology and it is difficult to have doubts in their ratio.

The modern technique permits directly to check the conception of Dobrovolsky zone.

Tsolis and Xenos in their paper devoted to the ionospheric precursors of earthquakes exactly before the L'Aquila earthquake

Tsolis G.S., Xenos T.D. A qualitative study of the seismo-ionospheric precursors prior to the 6 April 2009 earthquake in L'Aquila, Italy, Nat. Hazards Earth Syst. Sci. – 2010. – 10. – P. 133–137.

using the cross-correlation technology proposed by Pulinets et al.

Pulinets S.A., T.B. Gaivoronska, A. Leyva Contreras, L. Ciraolo, Correlation analysis technique revealing ionospheric precursors of earthquakes, Natural Hazards and Earth System Sciences, 4, pp. 697-702, 2004

demonstrated the validity of the Dobrovolsky zone conception (see Figure 1). They show the drop of cross-correlation coefficient between the stations inside of Dobrovolsky zone and no reaction for the configuration of stations outside the Dobrovolsky zone (red curve in the figure 2).

[Figure]

**Fig. 1.** The geographical location of Rome, San Vito and Athens the ionospheric stations, regarding the epicenter (blue triangle). Earthquake preparation area is plotted in red.

[Figure]

**Fig. 2.** Plot of the correlation coefficient of the denoised foF2 signals between Rome-Athens, Rome-San Vito, and San Vito Athens. Red arrow indicates the day of the seismic event and, black arrows represent ionospheric precursors.

The group of Valerio Tramutoli in the set of publications devoted to the TIR anomalies before the L'Aquila earthquake (again returning to Masci's comments that their no publications on L'Aquila precursors)

Genzano N., Aliano C., Corrado R., Filizzola C., Lisi M., Mazzeo G., Paciello R., Pergola N., Tramutoli V., RST analysis of MSG-SEVIRI TIR radiances at the time of the Abruzzo 6 April 2009 earthquake, Nat. Hazards Earth Syst. Sci., 9, 2073-2084, doi:10.5194/nhess-9-2073-2009, 2009

N. Pergola, C. Aliano, I. Coviello, C. Filizzola, N. Genzano, T. Lacava, M. Lisi, G. Mazzeo, and V. Tramutoli, Using RST approach and EOS-MODIS radiances for monitoring seismically active regions: a study on the 6 April 2009 Abruzzo earthquake, Nat. Hazards Earth Syst. Sci., 10, 239-249, doi:10.5194/nhess-10-239-2010, 2010

M. Lisi, C. Filizzola, N. Genzano, C. S. L. Grimaldi, T. Lacava, F. Marchese, G. Mazzeo, N. Pergola, and V. Tramutoli, A study on the Abruzzo 6 April 2009 earthquake by applying the RST approach to 15 years of AVHRR TIR observations, Nat. Hazards Earth Syst. Sci., 10, 395-406, doi:10.5194/nhess-10-395-2010, 2010

demonstrated the spatial distributions of TIR anomalies before L'Aquila earthquake from different satellite sources, and all of them have shown the Large scale TIR anomalies before the earthquake. Having in mind that there could be critics similar to Dr. Masci, they made analysis not only for the month

and year to L'Aquila earthquake but for other years demonstrating that the observed TIR anomalies a the real pre-earthquake anomalies.

Pulinets et al. and their presentations and publications

Pulinets S.A., Tramutoli V., Genzano N., Yudin I.A., TIR anomalies scaling using the earthquake preparation zone concept, 2013 AGU Meeting of the Americas, Cancun, Mexico, 14-17 May 2013, Paper NH42A-06

Pulinets, S.A., Ouzounov, D.P., Davidenko, D.V., Is Earthquake Forecasting Possible?! Integral Technologies of Multiparameter Monitoring over Geoeffective Phenomena in the Framework of the Complex Model of the Earth's Lithosphere–Atmosphere–Ionosphere Coupling, Moscow: Trovant, 2014.

provided the scaling of TIR anomalies using the Dobrovolsky $R = 10^{0.43}$ and Bowman et al. $R = 10^{0.44}$ estimations what id presented in the Figure 3

[Figure]

Fig. 3. TIR anomalies (yellow and red) registered before the L'Aquila earthquake (after Genzano et al., 2010). Blue circle – Dobrovolsky zone; red circle – Bowman et al. zone.

And now we return to the Masci's comment on Preparation zone size and epicenter determination. He is absolutely right that for the Tohoku earthquake the radius of preparation zone will be 7413 km. By the way it is confirmed by many experimental evidences. I can comment this by two arguments:

1. We provide mutimarameter monitoring of the different type of precursors. One of them give more precisely the position of epicenter, other give estimation of the magnitude, and the last – the time of event. Looking from perspective of short-term forecast – the most difficult is just the time of earthquake, and having in mind that the maximum intensity of anomalies is observed ~5 days before earthquake, the registering of anomaly of large scale give us at least estimation of time of earthquake. But here we come to the second point.

2. All of us learned elementary geometry in school, and determination of the circle center position should not present the great problem for majority of us regardless on the circle radius size. I would like to present such a procedure like in L'Aquila case but for the larger M7.8 Gujarat 2001 year earthquake for which Genzano et al. also measured the TIR anomalies:

Genzano, N., C. Aliano, C. Filizzola, N. Pergola, V. Tramutoli, A robust satellite technique for monitoring seismically active areas: The case of Bhuj–Gujarat earthquake, Tectonophysics. – 2007. – 431. – P. 197–210.

[Figure]

Fig. 4. TIR anomalies (yellow and red) registered before the Gujarat earthquake (after Genzano et al., 2007). Blue circle – Dobrovolsky zone; red circle – Bowman et al. zone.

So one can see that 7413 km for the Tohoku earthquake is reality. The problem of TIR anomalies is that thy depend of cloudiness and are not seen for every earthquake (is matter of lack). The determination of the circle center is also problem resolved using other type of precursors. For example. the ionospheric anomaly before the L'Aquila earthquake is "sitting" exactly over the epicenter (see Figure 5).

[Figure]

Fig. 5. GPS TEC differential map registered on 5 April 2009 using the data of Italian network of GPS receivers (Pulinets and Ouzounov 2016)

Pulinets S., Ouzounov D., Earthquake precursors in atmosphere and ionosphere. A review and future prospects, EGU 2016, Session NH4.7/AS4.37/EMRP4.21/SM3.5 - Short-term Earthquakes Forecast (StEF) and multi-parametric time-Dependent Assessment of Seismic Hazard (t-DASH), 2016

Concluding the first part of my comments I want firmly state that the Dobrovolsky et al. estimation of the earthquake preparation zone has the same order of fundamental meaning for seismology as the Gutenberg-Richter FMR relationship.

The second my concern from the Masci comments is that coincidence in time is not enough to classify the anomaly as a precursor. It could be so if to consider every anomaly individually. But we deal with the complex system approaching to the critical state what was perfectly demonstrated by Angelo de Santis et al:

De Santis A., Cianchini G., Favali P., Beranzoli L., Boschi E., The Gutenberg–Richter Law and Entropy of Earthquakes: Two Case Studies in Central Italy, *Bulletin of the Seismological Society of America*, 101, 1386–1395, 2011

Using the Shannon approach to the entropy estimation the authors calculated the period of approaching the system to critical state which coincides with the foreshock period determined by Papadopoulos (also for the L'Aquila case):

Papadopoulos G.A., Charalampakis M., Fokaefs A., Minadakis G., Strong foreshock signal preceding the L'Aquila (Italy) earthquake (Mw 6.3) of 6 April 2009, Natural Hazards and Earth System Sciences, 10, 19–24, 2010

But even more interesting fact is that the all geophysical anomalies detected before the L'Aquila earthquake (including those described in the reviewed paper) arose within the same time interval what

means that this is not simple coincidence in time and these anomalies are part of the general process of the development of the latest stage of the seismic cycle. All these processes are in state of synergetic interaction characteristic to the open systems, and this interaction is described in the paper:

Pulinets S.A., The synergy of earthquake precursors, Earthquake Science, 24, 535-548, 2011, doi:10.1007/s11589-011-0815-1

IT is possible to observe the anomalies propagation from ground up to the ionosphere presented in the Figure 6

[Figure]

Fig. 6. Temporal dynamics of radon release and variations of atmospheric and ionospheric parameters before the L'Aquila earthquake.

This interaction has also the physical background described in the paper:

Pulinets S.A., Ouzounov, D.P., Karelin A.V., Davidenko D.V., Physical Bases of the Generation of Short-Term Earthquake Precursors: A Complex Model of Ionization-Induced Geophysical Processes in the Lithosphere–Atmosphere–Ionosphere–Magnetosphere System, Geomagnetism and Aeronomy, 55, No.4, 540-558, 2015

and including the Coversphere in the set of interacting geophysical shells before earthquake is important contribution into this model.